

# Ca-rich garnets and associated symplectites in mafic peraluminous granulites from the Gföhl Nappe System, Austria

Konstantin Petrakakis[1], Nathalie Schuster-Bourgin[2], Gerlinde Habler[2], and Rainer Abart[2]

[1]University of Vienna, Department of Geodynamics and Sedimentology
[2]University of Vienna, Department of Lithospheric Research

**Correspondence:** Konstantin Petrakakis (konstantin.petrakakis@univie.ac.at)

**Abstract.** Mafic peraluminous granulites associated with the mantle-derived peridotites of the Dunkelsteiner Wald provide evidence of the tectono-metamortphic evolution of rocks in the Gföhl Nappe System, Austria. They contain the primary assemblage garnet + Al-rich-clinopyroxene + kyanite. Large Ca-and Mg-rich garnets are embedded in a granoblastic matrix of Al-rich-clinopyroxene, Ca-rich-plagioclase and minor hornblende. They have been partially replaced by different, locally con-

trolled symplectites of (a) corundum + sapphirine + spinel + plagioclase formed around kyanite inclusions, (b) orthopyroxene + spinel + plagioclase ± hornblende formed at their rims and (c) clinopyroxene + orthopyroxene + spinel + plagioclase ± hornblende formed within cracks. Garnets are built up from repeatedly occurring garnet types characterized by specific component distributions. Areal extend and cross-cutting relations allowed for the derivation of the relative timing of garnet types formation. Recrystallization and compositional readjustment of the reactive garnet volume during symplectite formation has led to

the development of pronounced, secondary diffusion-induced zoning profiles overprinting the different garnet types and postdating the complex garnet compositional structure. Thermodynamic analysis showed that none of the garnet types represents a preserved equilibrium composition. Furthermore, the latest garnet types show evidence of metasomatic (Fe+Mg)-loss affecting their Ca-content. The primary assemblage is stable between 760 and 880° C and pressures >11 kbar. The crack- symplectites are almost isochemical to the oldest garnet type and have been formed above 730° C and pressures between 7.5 and 5 kbar.

The studied rocks have undergone a more or less isothermal decompression from pressures above 11 kbar to ~6 kbar at temperatures about 800° C. Crack- and rim- symplectites have been formed after decompression during approximately isobaric cooling under conditions of low differential stress. Due to limited availability of fluids promoting symplectite formation, the time-scale of symplectite formation calculated from secondary diffusion profiles associated with crack- symplectites is shown to be geologically very short (< 0.5 ka).

## 1 Introduction

Over about the last three decades, the assessment of the *PT*- evolution of granulites and associated rocks within the Moldanubian Gföhl Nappe System is the focus of several tens of papers. The number alone of these papers points to a poor agreement of the estimated *PT* and *PT*-path interpretations. Table 1 is a representative selection of recent proposals.





The estimates are based on three lines of paragenetic analysis and/or *PT*- calculation. The first line used conventional one-reaction thermobarometry involving integral ternary feldspar compositions from perthite and antiperthite, Na-bearing clinopyroxene and the Ca-content in garnet (O'Brien, 2008; Vrána et al., 2013, and references therein). The calculated *PT*- conditions are about 1000° C and 15–20, occasionally more, kbar. The second line claims additionally the derivation of the prograde path of the rocks with the aid of equilibrium assemblage diagrams (pseudosections) mostly combined with results of conventional geothermobarometry (Štípská and Powell, 2005a; Racek et al., 2008; Štípská et al., 2014a; Jedlicka et al., 2015, 2017). This line implies the most complex geodynamic evolution of the rocks (cf.Table 1). The third line aims similarly at the derivation of *PT*-conditions, but points also to anatectic and open- system processes, that have affected rocks and minerals under the specific "Moldanubian" conditions (Hasalová et al., 2008b; Štípská et al., 2014a).

There is general agreement that the Moldanubian granulites have experienced at least decompression at high temperatures. The most probable reason for the observed divergence of *PT*- estimates is that in such rocks re-adjustment of rock microstructures, mineral abundances and compositions during their multi-stage tectono-metamorphic evolution may hinder the calculation of consistent *PT*- estimates. Critical discussions about the deviation of minerals from equilibrium composition are given in most of the papers in Table 1. The discussions are addressing the diffusion-aided Na-loss in clinopyroxene during the granulite facies overprint, the effects of decompression and rock recrystallization, and/or the available thermodynamic properties of minerals and their validity for thermobarometric calculations. All estimates however are based explicitly or implicitly on the assumption that the garnets and especially their Ca- contents have been hardly affected by processes other than changes of *PT* and that, therefore, Ca-zoning might have preserved past *PT* information.

In this paper, we address Ca-rich garnets and associated symplectites in mafic, garnet-clinopyroxene-kyanite granulites. Based on extensive microprobe analysis and element mapping as well as thermodynamic modeling of the component isopleths, it is inferred that primary zoning in garnet has been largely obliterated by intracrystalline diffusion and metasomatic alteration. Garnet composition modification has predated a largely isothermal decompression of the rock that started above 11 kbar at a temperature of c. 800° C. This decompression induced garnet break-down and formation of different, locally controlled symplectites within a very short time interval during early, post-decompression, $H_2O$-undersaturated conditions that were characterized by strain-free, almost isobaric cooling at c. 6 kbar.

Methods of data acquisition and recalculation (rock and mineral analyses) as well as of symplectite modal analysis, bulk composition calculation and EBSD of crack symplectites are given in the Supplement.

The type face used throughout the text is as follows: SOLID SOLUTION; *phase_component*; Pure_phase. Mineral and component abbreviations are as follows: CPX = clinopyroxene *di, hd, cTs, jd* [diopside, hedenbergite, Ca-Tschermak's pyroxene, jadeite]. GRT = garnet; *alm, sps, prp, grs* [almandine, spessartite, pyrope, grossularite]. HBL = hornblende; $X_{mg}$ [Mg/(Mg+Fe)]. OL = olivine; *fo, fa* [forsterite, fayalite]. OPX = orthopyroxene; *en, fs* [enstatite, ferrosilite]. ILM = ilmenite; $X_{mg}$ [Mg/(Mg+Fe)]. SPL = spinel; *spl, hrc* [spinel, hercynite]. SPR = sapphirine; $X_{mg}$ [Mg/(Mg+Fe)]. PL = plagioclase; *an, ab* [anorthite, albite]. Crn = corundum. Ky = kyanite. Qtz = quartz. Rt = rutile. Tnt = titanite. Additional normative components: *hy* = hypersthene; *ol* = olivine.



**Table 1.** Selection of proposed *PT* estimates from Moldanubian granulites and associated rocks in the Gföhl Nappe System, Moldanubian Unit. References: 1: Štípská and Powell (2005a). 2: Štípská and Powell (2005b). 3: Racek et al. (2006). 4: O'Brien (2008) and references therein. 5: Racek et al. (2008). 6: Faryad et al. (2010). 7: Vrána et al. (2013) and references therein. 8: Štípská et al. (2014a). 9: Jedlicka et al. (2015). 10: Jedlicka et al. (2017).

| P (kbar) \| T (°C) | | | | | Methods | Refe-rence |
|---|---|---|---|---|---|---|
| Initial stage | Implications | Peak | Implications | Post-Peak | | |
| | | 18 \| 850 | | | EAD | 1 |
| magmatic, dry @ 8-10 \| >1000 | → compression → | < 18 \| < 850 | → decompression → | 8-9 \| 800 | EAD | 2 |
| | | 15 \| 800 | | | EAD | 3 |
| 10 \| ? | | 15 \| ? | | 7-10 \| 750 | | 3 |
| | | 15-22 \| ~1000 | | | GTB | 4 |
| magmatic, @ 14 \| >950 | → compression → | 10-17 \| 650-850 or 20 \| 770 | → decompression → | 5-7 \| < 800 | EAD | 5 |
| | | 22 \| 900 | → decompression → | 16 \| 820 or 10 \| 870 | EAD | 6 |
| | | 22-23 \| ? | | | GTB | 7 |
| eclogite @ 18 \| 900 | → decompression & metasomatism to granulite → | 12 \| 950 | | | EAD+open-sys | 8 |
| 6 \| 400 | → prograde *PT* → | 32-40 \| ~700 | → decompression & re-heating → | 15-20 \| 800-1000 | EAD+open-sys | 9 |
| 8-9 \| 460 | → prograde *PT* → | 24-25 \| 550 | → decompression & re-heating → | 16-18 \| 800-870 | EAD | 10 |

**Methods:** EAD = Equilibrium assemblage diagrams (pseudosections); EAD+open-sys = EAD and additional consideration of open-system processes; GTB = conventional one-reaction geothermobarometry.

## 2   Geological setting and previous work

The Moldanubian Unit in Austria forms the eastern-most, belt-like part of the Variscan Orogen with a general NNE-SSW strike. In its western part, it has been intruded by granitic plutons of the South Bohemian Batholith; in the east, it is thrusted over rocks of the Moravo-Silesian Unit (Schulmann et al., 2008, Fig. 1). It is composed of three lithotectonic nappe systems that

are generally dipping to the east. The Gföhl Nappe System takes the highest tectonic position with felsic mylonitic granulites occurring at its top. The granulites are intimately underlain by the widespread Gföhl Gneiss and subordinate amphibolites. Garnet-peridotites with associated pyroxenites and eclogites represent tectonically emplaced fragments of the Earth's upper mantle (Medaris et al., 2005; Svojtka et al., 2016, and references therin) within the felsic granulites and rarely Gföhl Gneiss. The Drosendorf Nappe System (lithologically Bunte or Variegated Series) lies below the Gföhl Nappe System. It is composed

of amphibolites, marbles, gneisses and quartzites overlying the basal Dobra Gneiss. It is also exposed within the Gföhl Nappe System in the Drosendorf Window. The tectonically lowest Ostrong Nappe System (lithologically Monotonous Series) is in contact with the Southern Bohemian Batholite and is composed of migmatic, garnet-free Crd-gneiss with subordinate felsic gneiss and rare eclogitic bodies.

The metamorphic grade of the granulites within the Gföhl Nappe System has already been addressed in the previous section.

Hasalová et al. (2008a) studied the gneisses and migmatites of the Gföhl Nappe System showing structural transitions from





**Figure 1. Simplified geological map of the Austrian part of the Moldanubian Unit** modified after Schnabel (2002), Krenmayr et al. (2006), Cháb et al. (2007) and Kalvoda et al. (2008). DW signifies the Drosendorf Window. The inlay is a sketch of the sampling area of the mafic granulites (embedded within felsig granulites) at the steep flanks of Mitterbachgraben. The land road Gansbach-–Kicking as well as the GPS-coordinates at the star are shown (after Sheet 37, "Mautern", Geological Survey of Austria).





banded orthogneis to stromatic, schlieren and nebulitic migmatites formed by anatectic melt infiltration at conditions between < 850 °C and 7.5 kbar to 690 °C and 4.5 kbar. Anatexis is a widespread phenomenon within the Gföhl Nappe System affecting not only the Gföhl Gneiss, but also the mylonitic felsic granulites and amphibolites. This is most evident in many outcrops along the river Danube (Fig. 1). Anatectic conditions of 700–800 °C at c. 8 kbar and evidence for decompression has been

documented also for the Drosendorf Nappe System (Petrakakis, 1997; Racek et al., 2006). The Ostrong Nappe System differs distinctly from the other Moldanubian nappe systems. It shows similar anatectic temperatures of c. 720 °C, but at much lower pressures of >4.5 kbar. Based on rare relics of kyanite and staurolite, the rocks had attained a prograde maximum at <600 °C and c. 6 kbar before anatexis (Linner, 1996).

Seven samples of mafic granulites were collected from loose boulders dispersed over the steep flanks of the Mitterbachgraben
(inlay, Fig. 1). The local bedrock comprises serpentinites pertaining to the mantle-derived peridotite of the Dunkelsteiner Wald. The ultramafic rocks form several 100 m long lensoid bodies embedded in mylonitic felsic granulite. The collected samples are noticeable in the field, as they do not belong to the regionally widespread rock types in this area. They are dark gray, middle- to fine-grained, mostly granofelsic mafic granulites containing abundant pyroxene and kelyphitic reddish-brown garnets of occasionally striking large size up to 1.5 cm.

Previous work on similar rocks from the same area has been presented by Carswell et al. (1989). They interpreted the protoliths of these silica undersaturated, Mg-rich rocks showing high normative plagioclase contents as cumulate-liquid mixes of intra-plate basaltic magmas with lower crust rocks. The observed primary metamorphic assemblage GRT+CPX+Ky should have been formed at 30 kbar and 1146–1102 °C. Rarely observed SPR+PL- symplectites were formed at 1000 °C and $P$ < 20 kbar at the expense of Ky+GRT. Finally, hornblende rimming matrix clinopyroxene was formed at amphibolite facies
conditions.

XRF-analyses of the collected samples reveal K-poor, Mg-rich compositions with $X_{mg}$ ranging within 0.70–0.82 (see Supplement). In terms of normative contents, they contain *crn* in the range 10 to 16 %. Three of them contain *an* in the range 9 to 19 %; the other four samples contain instead *ol* between 3 and 9 %. The *di* contents of all samples vary between 48 and 61% and the *hy* content between 3 and 19 %. The *ab* content varies between 9 and 19 %. Some of the samples (UM5, UM6, UM8)
resemble the corundum-bearing garnet clinopyroxenites from the Beni Bousera ultramafic massif that have been considered as low pressure crystallization cumulates from plagioclase-rich gabbros of ophiolitic affinity that underwent subduction and re-equilibration at mantle conditions (Kornprobst et al., 1990). Svojtka et al. (2016) assigned the Dunkelsteiner Wald pyroxenites to LREE-enriched melts of the subcontinental lithospheric mantle. In comparison with $Al_2O_3$ contents of 15–24 wt.-% in the samples presented here, their pyroxenitic samples are characterized by significantly lower $Al_2O_3$ not exceeding 12.23
wt-% and higher $X_{mg}$ = 0.87 – 0.90. Based on the pronounced peraluminous composition variability, we consider our samples as mantle-derived clinopyroxenitic melts that have assimilated variable amounts of Al-rich crustal material during ascent and tectonic emplacement to their current position.

The following presentation and discussion is focused on sample UM8 that contains some large garnets with an unusual high number of features. This sample is the most magnesian and peraluminous of the whole collection ($X_{mg}$ = 0.82 with normative
*crn* = 14.07 % and *an* = 18.78 %).





## 3   Rock and mineral features

The most abundant phase in the mafic granulite UM8 is clinopyroxene followed by plagioclase, fewer garnet of occasionally large size and still fewer brownish hornblende (Fig. 2a). The rock matrix is fine-grained with typical crystal sizes ranging from 0.1 to 0.3 mm. Beside clinopyroxene and plagioclase, sporadic relics of garnet associated with orthopyroxene- bearing
symplectitic intergrowths occur within the rock matrix (Fig. 2b). Hornblende is common, but a minor phase (cf. Fig. 2c). Fine-grained hornblende and white mica are abundant along the conspicuous darker, thin and straight bands that cross-cut the rock matrix (Fig. 2a).

Under the light polarizing microscope, the matrix is granoblastic, almost equigranular, comprising smooth to straight inter-faces among clinopyroxene, plagioclase and hornblende that meet at triple junctions (Fig. 2c). The large, up to 1.3 cm sized
garnets (Fig. 2a) are partially kelyphitized and contain kyanite and clinopyroxene inclusions (Fig. 2d). Plagioclase is often observed with clinopyroxene in the inclusion domains of garnet, but, as will be discussed later, its origin is secondary. Kyanite inclusions have been partially to completely replaced by symplectites of plagioclase and tiny, highly refractive minerals (Fig. 2d). Granular green spinel may be occasionally recognized among them. In addition, these garnets exhibit 0.1–0.2 mm wide mode-I (extensional) cracks, which are filled with fine-grained symplectite (Fig. 2e). Most garnets are however much smaller
with sizes less than 4 mm (Fig. 2f). Such garnets are kelyphitized too, but kyanite inclusions are seldom. They typically contain only clinopyroxene and plagioclase in their inclusion domains.

The garnets of the rock have been partially replaced by different symplectites. Based on their formation environment, three types of symplectites can be discerned:

*Symplectites around kyanite inclusions in garnet* (Fig. 3). The primary assemblage GRT+Ky broke-down forming initially
a coarse-grained symplectite of SPL+PL+Crn (Fig. 3a,b). Corundum intergrown with spinel is then overgrown and replaced partially by sapphirine (Fig. 3c). Finally, the initial SPL+PL+Crn- symplectite has been partially replaced by a fine-grained SPR+PL- symplectite. Noticeably and as shown in Fig. 3d and Fig. 3e, the break-down products corundum and sapphirine are intergrown with the retreating garnet edge. GRT+Ky break-down has led elsewhere to the complete consumption of kyanite (Fig. 3f). The contrast-enhanced enlargement in Fig. 3g shows that also sapphirine has been overgrown by spinel belonging to
an extremely fine-grained OPX+SPL- symplectite. This symplectite presumably replaced partially the coarse-grained SPR+PL-symplectite. The whole symplectitic structure is enclosed in a plagioclase corona, which shows straight contacts with the breaking-down garnet. All these features suggest that the current picture of the inclusion-related symplectites evolved during a multi-stage transformation process, whereby the initially formed corundum and the descendant sapphirine became successively metastable. This type of symplectite is henceforth referred to as *inclusion-related symplectite*.

*Symplectites at garnet rims* (Fig. 4a, Fig. 4b). This kind of symplectite comprises the observed kelyphites formed by reaction between the garnet rim and the rock matrix. It shows best the typical features of a symplectite microstructure. It consists of intergrowths of vermicular crystals replacing the garnet along a sharp garnet–symplectite reaction front. The older parts of the symplectite are distant to the garnet and comprise coarser-grained, rather granoblastic [OPX+SPL]- domains (Obata, 2011). Towards the garnet, the symplectitic phases become younger, reducing their intergranular spacing (and size), and evolving to



**Figure 2.** (Caption in the next page)





an intergrowth of fibrous crystals of OPX+PL increasingly aligned perpendicular to the garnet–symplectite interface (Obata, 2011, „Law of normality"; Fig. 4a). Hornblende is often observed within this kind of symplectites. It is either intergrown with plagioclase touching the reaction front and being a late symplectitic phase (Fig. 4b), or forms coarse-grained granular crystals distant to the reaction front (Fig. 4a). As shown in both figures, the outermost margins of the adjacent matrix plagioclase

crystals are enriched in Ca. This type of symplectite is henceforth referred to as *rim-symplectite*.

*Symplectites within Mode-I (extensional) garnet cracks* (Fig. 2e, Fig. 4c, Fig. 4d). This type of symplectite shows a symmetrical shape and size distribution of the phases on either side of the crack middle-line. Along the middle-line, coarser-grained orthopyroxene- and clinopyroxene-domains are intergrown with tiny vermicular spinel. Towards both garnet–symplectite interfaces, they evolved to fine-grained intergrowths of fibrous OPX+PL±CPX crystals oriented perpendicular to the reaction

front. Thereby, inter-granular spacing (and size) of the symplectitic phases decreases. Some few hornblendes are often present and follow the modes of occurrence described in rim- symplectites. This type of symplectite is referred to below as *crack-symplectite*.

All three types of symplectites have been observed at one large garnet within one thin section. Rim- and crack- symplectites are mineralogically similar, bearing the assemblage [OPX ± CPX+SPL]+PL. Clinopyroxene occurs however only in crack-

symplectites. Inherent to the formation of all three types of symplectite are the secondary compositional changes within some tens of $\mu$m across the retreating garnet edge. This is addressed in the next section.

## 4   Mineral compositions

The mineral compositions (microprobe analyses) addressed below are given in the Supplement.

---

**Figure 2. Rock and mineral features in sample UM8**. Figures (a), (b), (d) and (e) are taken with a polarizing microscope. **(a)** Thin section showing one of the largest, partially kelyphitized garnets ($\oslash$ ~1.3 cm). The abundant gray crystals in the matrix are clinopyroxene with scarce garnet relics; white crystals are feldspar and the light brownish ones hornblende. The thin straight darker bands cross-cutting the rock matrix are rich in hornblende and white mica. The black cycles are orientation marks for microprobe analysis. **(b)** Granoblastic rock matrix showing clinopyroxene and plagioclase crystals with subordinate, small brownish crystals of hornblende. The small, irregular, highly refractive constituents are relics of garnet and symplectitic intergrowths containing orthopyroxene. **(c)** BSE image of the rock matrix showing smooth interfaces and triple junctions among clinopyroxene, plagioclase and hornblende as well as thin exsolution lamellas in the clinopyroxene interior. The plagioclase rims are enriched with Ca. **(d)** Kyanite and clinopyroxene inclusions in garnet partially or totally replaced by symplectite of plagioclase and highly refractive minerals. **(e)** 0.1—0.2 mm wide, mode-I cracks in garnet filled with fine-grained symplectite. **(f)** BSE image of a middle-sized garnet showing an inclusion-related symplectite within the resorption garnet embayment and a kelyphite at rims. **(g)** Ca-distribution map of the garnet in figure (a). The lines Z1-Z2-Z3, X-Y, P1-P2 and Q-S, T-Y, and the areas designated with yellowish boxes refer to the profiles and GRT-types (see text), respectively, shown in Fig. 5. The dashed line delimits the distribution of GRT-type C; the dotted line delimits the distribution of GRT-type Z1. GRT-type Z3 (highest Ca content) is restricted along the lower right margin of the garnet. The rectangle named SG shows the area, where garnet subgrains as those in Fig. 7 occur.



**Figure 3. BSE images of inclusion- related symplectites. (a)** Relict kyanite inclusion in garnet surrounded by Crn (black) + PL(middle gray)- symplectite. The line X-Y refers to the profile in Fig. 5c and Fig. 2g. The rectangles B and D refer to the figures (b) and (d), respectively. **(b)** Crn+PL- symplectite seemingly enclosing an extremely fine-grained SPR+PL- symplectite. **(c)** In fact, sapphirine has overgrown and consumed corundum that is intergrown with spinel. **(d)** Corundum intergrown with the reactive garnet edge. Note the late hydration products (white mica and prehnite) as well as a late crack filled with albite. **(e)** Sapphirine has overgrown the resorbing kyanite and is intergrown with the reactive garnet edge. **(f)** SPR+PL- symplectite in garnet (cf. Fig. 2f ). A plagioclase corona isolates the symplectite from the garnet. Note that here kyanite is completely consumed. On the lower right part of this figure a typical garnet-rim- symplectite (cf. Fig. 4a) is shown. The rectangle G refers to figure (g). **(g)** An extremely fine-grained symplectite of PL+OPX+SPL is seemingly enclosed in the coarse-grained SPR+PL- symplectite. Note however, that spinel has overgrown sapphirine.





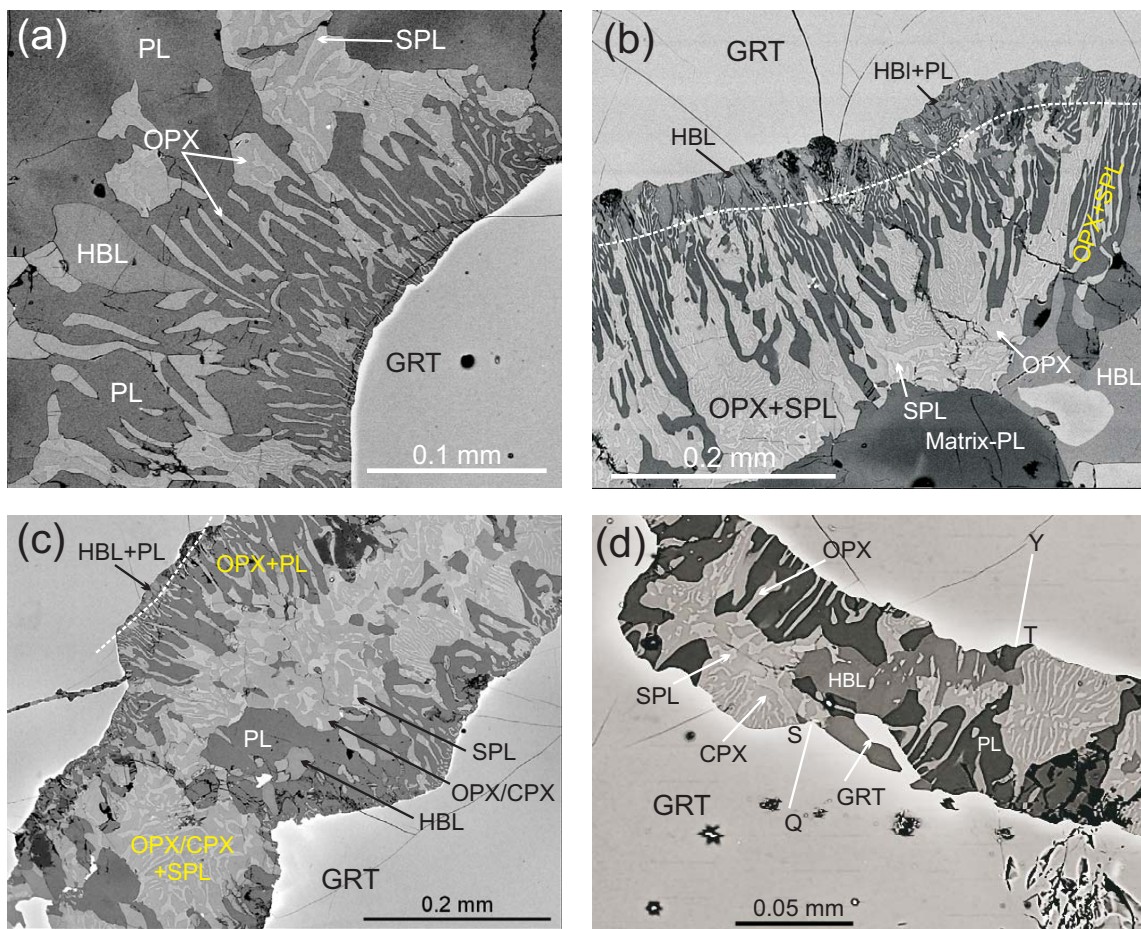

**Figure 4. BSE images of rim- and crack- symplectites.** Note in all figures the reduction of spacing (and size) of the symplectitic phases from the rock matrix [figures (a) and (b)] or the garnet crack middle-line [figures (c) and (d)] towards the garnet–symplectite interfaces. Note also the light-colored narrow zones at the retreating garnet edges denoting the overprinted secondary diffusion profiles (see text) associated with the symplectite formation. **(a)** OPX+SPL+PL- symplectite forming the kelyphites at garnet rims. At the left upper part of the figure, the matrix plagioclase crystals show a zonal pattern resulting from Ca-enrichment of their margins. **(b)** Composite garnet kelyphite showing a OPX+SPL+PL-symplectite adjacent to rock matrix evolving to a HBL+PL- symplectite touching the garnet–symplectite interface. The approximate boundary between the two symplectites is emphasized by a dashed line. In the lower central part of the figure, the outer-most rims of coarse-grained matrix plagioclase show light-colored margins due to Ca-enrichment. **(c)** Symplectite within a mode-I (extensional) garnet crack showing a symmetric arrangement of the symplectitic phases about the middle-line of the crack. At the upper left garnet edge, an OPX+PL- symplectite has evolved to a HBL+PL- symplectite over very narrow zone. The approximate boundary between them is emphasized by a dashed line. **(d)** CPX+OPX+SPL+PL- symplectite within a garnet crack. A clinopyroxene-domain intergrown with tiny vermicular spinel strictly oriented perpendicular to the garnet–symplectite interface as well as a plagioclase domain intergrown with vermicular orthopyroxene take up the largest area of the symplectite. Note also the remnant garnet within the kelyphite as well as the hornblende that shows a different mode of occurrence resulting most likely by late replacement of orthopyroxene. The garnet profiles along the lines Q-S and T-Y is shown in Fig. 5e.





**Figure 5.** Caption in the next page.



**Table 2. Composition and features of garnet types**. m (n) and ±s are the average and standard deviation of n consecutive point analyses of uniform composition along the garnet profile Z1-Z2-Z3 in Fig. 5a, see also Supplement, Table S 1.

| GRT-type: | Z1 | | Z2 | | C | | Z3 | | E | | Z3 - C |
|---|---|---|---|---|---|---|---|---|---|---|---|
| | m (50) | ±s | m (36) | ±s | m (32) | ±s | m (21) | ±s | m (18) | ±s | |
| $X_{alm}$ | 0.23 | 0.00 | 0.24 | 0.00 | 0.26 | 0.00 | 0.19 | 0.00 | 0.18 | 0.00 | -0.07 |
| $X_{prp}$ | 0.58 | 0.01 | 0.51 | 0.00 | 0.54 | 0.00 | 0.47 | 0.00 | 0.60 | 0.01 | -0.06 |
| $X_{grs}$ | 0.18 | 0.01 | 0.24 | 0.00 | 0.19 | 0.00 | 0.32 | 0.01 | 0.19 | 0.01 | 0.13 |
| Features overview: | $grs < alm < prp$; Occurrence ≠ Type C | | Lowest $prp$; $alm = grs$ | | $grs < alm < prp$; Occurrence ≠ Type Z1 | | Highest $grs$; $alm < grs < prp$ | | Highest $prp$; $grs = alm$ | | |

Beside the different types of symplectites, the largest garnets show systematic composition patterns. Fig. 5a shows the element distribution profile Z1-Z2-Z3 across the garnet in Fig. 2g. The Mn content in the garnets is generally very low. Mg, Ca and Fe vary along the profile, but Fe shows the lowest variation. Along the profile, distinct element distribution patterns occur that may be observed repeatedly in all other garnets of the rock (e.g. Fig. 5b,c,d,e). These patterns, addressed in the following

5 as GRT-types Z1, Z2, C, Z3, are discerned on the basis of distinctly differing relations among $X_{alm}$, $X_{prp}$ and $X_{grs}$ (Table 2). GRT-types Z1 and C show similar compositions characterized by $X_{grs} < X_{alm} < X_{prp}$. However, they differ distinctly in their mode of occurrence, see below. GRT-type Z2 is characterized by lowest $X_{prp}$ and $X_{alm} \approx X_{grs}$. GRT-type Z3 shows the highest observed $X_{grs}$ and $X_{alm} < X_{grs}$. The specific averaged values ($X_{alm}$; $X_{prp}$; $X_{grs}$) of these GRT-types are placed within yellowish boxes in Fig. 5a. Where appropriate, these boxes are included for reference in all other garnet profiles discussed below.

10 With the aid of the GRT-types defined as in Fig. 5a, the Ca-distribution map in Fig. 2g allows for mapping of their areal extend. The dashed line in Fig. 2g delineates the distribution of GRT-type C. The dotted line delineates the distribution of GRT-type Z1. The distribution of GRT-type Z3 is easily recognized by the high Ca-content (light gray color in the BSE image) at the right and lower margin of the garnet. GRT-type Z2 takes up the remaining part of the garnet.

---

**Figure 5. Composition of garnet. (a)** Profile Z1-Z2-Z3 in the large garnet of Fig. 2g. The specific compositional patterns occurring along this profile are standardized to GRT-types designated by yellow boxes (see text and Fig. 2). These GRT-types are repeatedly observed in other garnets of the sample, see the following figures. The yellow boxes are given there for reference. **(b)** Profile across the middle-sized garnet of GRT-type Z3 shown in Fig. 2e. The GRT-type E occurs adjacent to clinopyroxene inclusions. Note the pronounced zoning in Al shown by the clinopyroxene inclusion. **(c)** The diffusion profiles X-Y and P1-P2 may be located in Fig. 2g. They overprint GRT-type C across complex, inclusion-related symplectites, see text and Fig. 3. **(d)** Profile of a garnet of type C towards a rim- symplectite. Contrary to Ca that remains unaffected, the diffusion curves of the other elements show an inversion point at ~8 $\mu$m ahead the garnet edge. **(e)** The shown two profiles can be located in Fig. 2g. The curves are model diffusion profiles fitted at the analysis points by treating *alm*, *prp*, *grs* as independent components. Analyses of garnet in contact with the symplectite (contact garnet) and the seemingly unaffected garnet apart of the diffusion profile (initial garnet) are given in the Supplement, Table S2.




GRT-type C occupies the large, inclusion-poor interior part of the garnet. GRT-type Z1 has evolved at a strongly Ca-depleted area along a garnet crack, which can be recognized in Fig. 2a,g. Therefrom, it "intrudes" irregularly the garnet interior and extends over a narrow zone along the lowest rim of the garnet (Fig. 5g). As can be recognized in Fig. 2g, GRT-type Z1 cross-cuts type Z3 over a narrow transitional zone and is therefore younger. This age relation is supported also by the typical

middle-sized garnet in Fig. 5b. This garnet is of type Z3, but has evolved to GRT-type Z1 towards its margin. Compared with the other GRT-types shown in Fig. 2g, GRT-type Z1 is a late feature related most probably with the action of metasomatizing agents. GRT-types Z2 and C are seemingly older, but their temporal interrelation is not clear. Their transition towards GRT-type Z3 is smooth. Despite the compositional similarities, we discriminate GRT-type C from GRT-type Z1 based on their different modes of occurrence. Based on these observations, the relative age relations of the garnet types may be summarized as follows:

Z1 < Z3 < Z2 ≤ C. It is noteworthy that the whole garnet is keliphitized along the rim (cf. Fig. 2g) and, as shown by the dotted line at the lowest garnet rim (Fig. 5a), only the younger GRT-type Z1 may be formed as late as the rim- symplectite.

The garnet profiles in Fig. 5 show pronounced compositional zoning profiles that have evolved within some tens of microm-eters along the garnet interface with the symplectites. Such compositional zoning profiles within the reactive and retreating garnet edge are imposed over pre-existing garnet compositional patterns and are, therefore, secondary. As will be discussed

later, their evolution is linked to the formation of the symplectites. Henceforth they will be referred to as *diffusion profiles*.

A close inspection of Fig. 5 reveals that the diffusion profiles overprint the GRT-types Z2, C and Z3. Therefore, the associ-ated symplectites are younger than the overprinted GRT-types. For example, the strong Ca-depletion at 6 mm adjacent to an inclusion-related symplectite (Fig. 5a) is imposed over GRT-type Z2. The same kind of depletion at 10.5 mm is imposed over GRT-type C. In Fig. 5b, the same element depletion is overprinted over GRT-type Z3. Such relations can be verified also in

Figs. 5c,d and e. As such overprinting relations have not been observed in case of the younger GRT-type Z1, we conclude that symplectite formation is at earliest coeval to this garnet type and consequently younger than the internal structure of the garnet related to GRT-types Z2, Z3 and C.

The garnet profiles shown in Figs. 5c,e are acquired over domains of inclusion-related and crack-symplectites, respectively. The former shows a diffusion profile within about 10 $\mu$m towards the garnet–symplectite interface characterized by increasing

$X_{prp}$ and $X_{alm}$, and sharply decreasing $X_{grs}$. The latter diffusion profile is characterized by unchanged $X_{grs}$, increasing $X_{alm}$ and decreasing $X_{prp}$. This striking difference reflects primarily the influence of the local environment on their formation mechanism. In the former case, the environment is defined by GRT-type Z2 reacting with its kyanite inclusions. In the latter case, it is defined by the instability of GRT-type C alone. Diffusion profiles at rim- symplectites of GRT-type C as the one in Fig. 5d are similar to those of crack-symplectites. Occasionally and as shown in Fig. 5d, the diffusion curves for Mg, Fe, Mn adjacent to rim-

symplectites show an inflection point at some short distance before the retreating garnet edge.

Pronounced diffusion profiles in garnet are also observed towards clinopyroxene inclusion domains. As shown in Fig. 5b, the middle-sized garnet of GRT-type Z3 contains also GRT-type E in the vicinity of clinopyroxene inclusions. This type is characterized by the highest observed $X_{prp}$. Remarkably, one of the clinopyroxene inclusions crossed by the profile in Fig. 5b shows a sharp zonal pattern with highest Al-content in its core. Even for this seemingly "protected" inclusion, $X_{Na}$ remained

negligibly low. This clinopyroxene zoning is rather the exception. As shown in Fig. 6a, in contrast to the adjacent reactive garnet





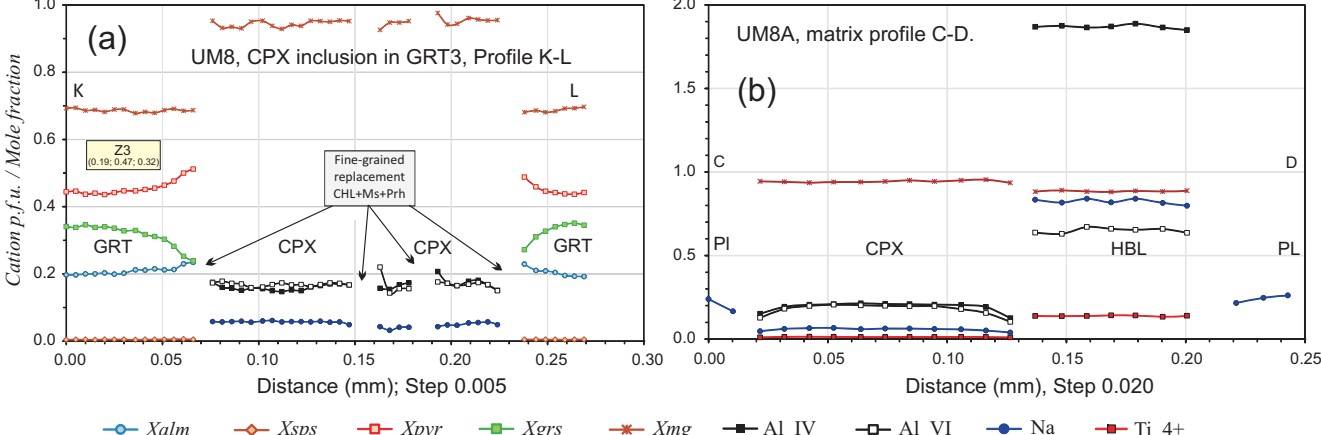

**Figure 6. Clinopyroxene composition.** (a) Profile K-L across a clinopyroxene inclusion in garnet. Contrary to the homogenized compositional pattern of the clinopyroxene, the adjacent garnet preserves a diffusion profile within ~30 $\mu$m towards the pyroxene. Note also the appreciable Al ($X_{cTs}$ ~0.20) and very low Na contents of the clinopyroxene. (b) Profile C-D (cf. Fig. 2c) across matrix phases. Clinopyroxene and hornblende are completely homogenized by intracrystalline diffusion. The composition of the clinopyroxene interior is similar with that of the inclusion shown in figure (a). Its outer-most rim shows less Al ($X_{cTs}$ ~0.12) correlating negatively with the outer-most rim of plagioclase showing $X_{an}$ ~0.85.

edges preserving diffusion profiles, clinopyroxene inclusions show a uniform element distributions ($X_{cTs}$ ~0.20; $X_{Na}$ ~0.07). In fact, the composition of clinopyroxene inclusions does not differ significantly from the composition of the clinopyroxene crystals in the rock matrix (Supplement, Table S3).

Figure 6b shows a profile across matrix crystals of plagioclase, clinopyroxene and hornblende. The element distribution within hornblende and clinopyroxene are perfectly uniform. This is interpreted as evidence of homogenization by intra crystalline diffusion. The matrix clinopyroxene shows a similar composition as the inclusion discussed above. Only its outer-most rim shows less Al ($X_{cTs}$ ~0.13, $X_{Na}$ ~0.05). This change correlates with the Ca-enrichment of the outer-most plagioclase rims (cf. Fig. 2c). As shown in Fig. 2c, the interiors of some larger matrix clinopyroxene crystals contain very thin exsolution lamellae. Notably, the average of 120 point analyses carried out with defocused beam in scanning mode over such an area of size 49×49 $\mu$m$^2$ does not differ significantly from the interior analysis of clinopyroxene.

Plagioclase in inclusion domains in garnet contain more Al ($X_{an}$ ~0.88) than the matrix crystals. The latter show interiors with uniform composition of $X_{an}$ ~0.72 and outer-most rims of $X_{an}$ ~0.84 (Fig. 2c; Supplement, Table S4). Plagioclase in inclusion domains of garnet has been often partially or totally replaced pseudomorphically by prehnite. Very late hydration and replacement products composed mostly of fine-grained aggregates of chlorite and sericite occur in almost all inclusion domains in garnet.




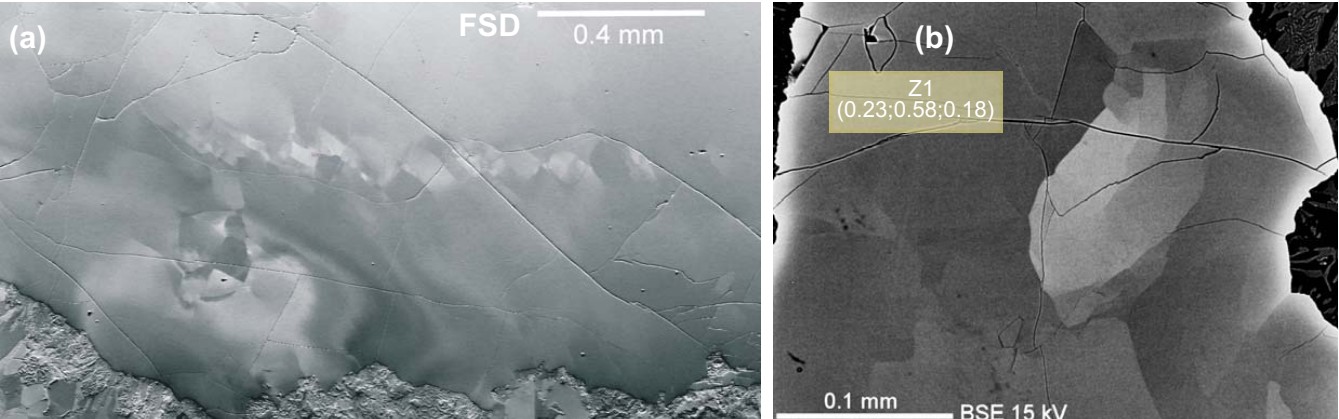

**Figure 7. Images of orientation contrast of garnet subgrains within garnet of GRT-type Z1. (a)** Garnet subgrains aligned linearly and forming an imbrication microstructure. **(b)** Subgrain development within garnet. The secondary diffusion profile at the right garnet rim (thin light-colored zone) in contact with a rim-symplectite has overprinted the subgrains- containing garnet.

## 5 Additional features of garnet

Some large garnets show grain-internal domains with pronounced contrast on the BSE images in compositionally very homogeneous areas (Fig. 7). Forward scatter electron (FSE) images as well as orientation imaging using EBSD revealed that the observed contrast is due to the presence of subgrains with up to 12° mutual misorientation. The subgrains are related by rota-
tion around a common axis coinciding with garnet [211]. Similar subgrain formation ascribed to intracrystalline deformation by dislocation glide has already been observed in granulites and upper mantle rocks (Kleinschrodt and McGrew, 2000; Kleinschrodt and Duyster, 2002). In the sample at hand, this deformation-induced feature can be observed only within the youngest GRT-type Z1. Notably, the diffusion profile (light-colored zone) in contact with a rim-symplectite at the right garnet rim (Fig. 7b), has overprinted the subgrain-containing garnet rim. Therefore, formation of the garnet subgrains within GRT-type Z1
pre-dates the crack- symplectites.

Some other garnets exhibit a peculiar appearance resembling a poikiloblastic microstructure (Fig. 8). We use the designation "poikiloblastic" for these garnets without any genetic co-notation. They contain abundant large, several tens to about two hundred $\mu$m sized, rounded to vermicular plagioclase inclusions. Such garnets resemble closely GRT-type Z1 and occur in the outermost 100 to 300 micrometers of large garnets (Fig. 8a) or throughout small, up to about a millimeter sized garnets in the
rock matrix (Fig. 8b). As can be recognized in the Fe-distribution map in Fig. 8a, the poikiloblastic garnet is overprinted by the diffusion profile that is associated with the crack- symplectite shown in the upper right part of the image. The poikiloblastic garnet is thus older than the crack- symplectite. The plagioclase inclusions show uniform chemical composition, but differ significantly between the two poikiloblastic garnets shown in Fig. 8.

Modal analysis of the phases in crack-symplectites (Fig. 4d, Fig. 5e) combined with their chemical composition showed that
the bulk symplectite composition is almost isochemical with the GRT-type C. GRT-type extends beyond the diffusion profile





**Figure 8. The poikiloblastic garnet.** The designation of these garnets is based on their resemblance with the common poikiloblastic microstructure. It is used without any genetic co-notation. **(a)** Ca, Fe and Mg distribution maps at the rim of a large garnet evolving towards a poikiloblastic garnet of GRT-type Z1. The dashed lines delineate the poikiloblastic garnet. At its upper right part signified with a double yellow arrow, a symmetrical crack-symplectite containing an [OPX+SPL]- domain cross-cuts the poikiloblastic garnet. The associated pronounced diffusion profile (best seen in the Fe- distribution map) overprints the poikiloblastic garnet. **(b)** BSE image of an poikiloblastic garnet in the rock matrix. The inclusion domain in the rectangle shows a diffusion profile around clinopyroxene, orthopyroxene and spinel inclusions. **(c)** Profile across the garnet in figure (b). The garnet close to the plagioclase inclusions shows a homogenized element distribution resembling that of GRT-type Z1. Close to the inclusion domain shown in Figure (b), the element distribution resembles rather that of the older GRT-type Z2.

shown in Fig. 5e acquired over a crack-symplectite. An analysis practically identical with GRT-type C and lying on this profile is #974 (Supplement, Table S3). The tools and methods of volumetric measurement of the symplectitic phases as well as the calculation of the bulk symplectite composition are described in the Supplement. The results are summarized in Fig. 9. They confirm that within the maximal uncertainty of lesser than 3 % shown by FeO, the integrated bulk symplectite is isochemical to





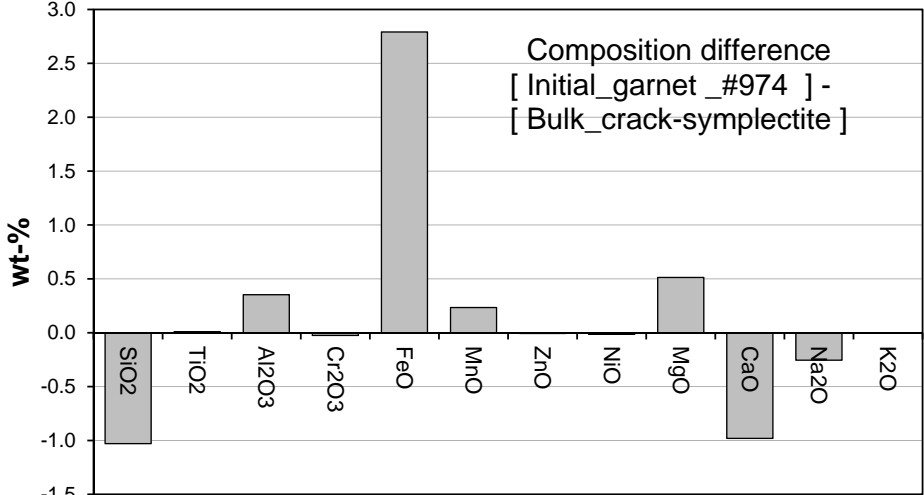

**Figure 9. Composition difference between the initial garnet and the bulk crack-symplectite**. See also Fig. 5e and Fig. 4d.

initial garnet #974. However, some minor input of $SiO_2$, $Na_2O$ and CaO to the bulk symplectite as well as some loss of $Al_2O_3$, FeO, MnO, and MgO from garnet should have taken place. Most likely, this mass exchange has been mediated by fluids. Despite these minor uncertainties, we consider the bulk symplectite as isochemical with #974 ( or GRT-type C) allowing thus for a straightforward thermodynamic modeling of the symplectite formation within a simple bulk chemical system defined by the composition of this garnet. This is presented next.

## 6  *PT*- evolution

Despite the large variability of the observed features, two of them serve as staring points for modeling the *PT* evolution of the rock. The first one concerns the primary assemblage GRT+CPX+Ky. The composition space for this assemblage is the rock bulk analysis. The second one is the crack- symplectite assemblage CPX+OPX+SPL+PL that is isochemical to GRT-type C. These assemblages fix two different stability fields that constrain the *PT* path. Although hornblende is occasionally present in small amounts in the inclusion domains of garnet and within some symplectites, we consider its formation as late due to extensive recrystallization and partial re-hydration of the rock matrix that has been set up during or after the latest stage of symplectite formation. This is discussed in detail in a following section. Therefore, the two assemblages above are considered free of primary hydrated minerals and independent of $a_{H2O}$ at the time of their formation.

For the calculation of the stability fields of the assemblages GRT+CPX+Ky and CPX+OPX+ SPL+PL, the Theriak— Domino software (de Capitani & Petrakakis, 2010), version 03.01.2012 with the associated database file jun92.bs was used. Because of the appreciable amounts of the *cTs* component in the measured clinopyroxene and the lack of appropriate *a-x* relations, a three-site ideal mixing model for clinopyroxene comprising the phase components *di, hd, cTs* and *jd* was formulated. Sapphirine was omitted, as the available data seem to be inappropriate for the peculiar composition of the rock sample. There-



fore, some parts of the following diagrams between the stability fields of the above two assemblages may be metastable with respect to sapphirine. Because of the very low, almost negligible Mn content in garnet and bulk rock, we added the minor Mn to Fe and utilized the ternary mixing model for garnet included in the database. We also omitted the very low contents of K measured in the rock and the minerals as well as P.

The equilibrium assemblage *PT* diagram (pseudosection) of Fig. 10a is based on the rock bulk composition. It reproduces well the two target assemblages mentioned above. The inlet in Fig. 10b shows the same diagram expanded up to 20 kbar. The primary assemblage GRT+CPX+Ky is stable at *P* > 11 kbar and *T* between 760 and 880 °C. From there, the stability field of the crack-symplectite CPX+OPX+SPL+PL may be reached by a more or less isothermal decompression to pressures of less than 9 kbar, but higher than c. 5 kbar. This low *P*-limit is constrained by the absence of olivine in the symplectites.

Figure 10b shows the isopleths of garnet equilibrium composition. If an original growth zoning acquired during propagation of the garnet growth front at local equilibrium has been preserved without any modification by intracrystalline diffusion, all component isopleths should intersect at a single *PT*- point. Based on this figure, Fig. 10c shows the isopleths distribution for the observed GRT-types. For none of the GRT-types does a common intersection point of the isopleths for the measured $X_{alm}$, $X_{prp}$ and $X_{grs}$ exists, indicating that none of the GRT-types corresponds to a preserved equilibrium state. This holds not only

for the stability field of the primary assemblage, but also for all other assemblages shown in Fig. 10a. This indicates that the garnet composition has been modified substantially during the evolution of the rock. Fe appears to have been partially lost and homogenized towards lower values for all garnet types. In case of GRT-types Z3 and E, the measured $X_{alm}$ isopleth lies far beyond the right side of the figure. The *prp* isopleths are much lesser dispersed and lie closest to the stability field of the primary assemblage. Ca seems to "resist" extensive modification, perhaps "memorizing" some prograde *PT* past of the rock,

especially within the Ca-rich GRT-type Z3. Taking however into account the relative timing of the garnet types (Z1 < Z3 < Z2 ≤ C), it becomes evident that, for example, the Ca-rich GRT-type Z3 cannot reflect some earlier preserved *PT* information. In this case, GRT-type Z3 should be the oldest compositional element of the complex garnet structure. Indeed, if Fe has been partially removed from the garnet (cf.also Fig. 9) pushing the measured $X_{alm}$ far to the right of Fig. 10c, the contents of Ca and Mg in garnet should have increased due to stoichiometric constrains. This increase should have displaced the isopleth of the

measured $X_{grs}$ to the left and the isopleth for the measured $X_{prp}$ to the right of this figure. The opposite is certainly also true. Therefore, the measured higher Ca contents in some GRT-types do not necessarily reflect preserved past *PT* information.

Evaluating the degree of deviation of the various GRT-type isopleths from equilibrium by their proximity to the stability field of the primary assemblage, it may be concluded that GRT-type C occurring in the inner part of the garnet is the least disturbed type with its *grs* isopleth still lying within this stability field. We consider therefore GRT-type C as the best available garnet

composition approaching better the stability field of the primary assemblage.

The measured compositions of inclusions in garnet are compared with calculated compositions along the decompression path shown in Fig. 10d. The pressure range 15 to 6 kbar at 825 °C is thought to be representative for the decompression implied by Fig. 10a. With the exception of garnet for *P* < 12 kbar, the calculated equilibrium composition of the minerals are rather insensitive to *P* change. The *cTs* and *jd* contents in clinopyroxene are predicted to decrease slightly with decreasing

pressure. Plagioclase is stable below ~12 kbar outside the stability field of the primary assemblage. Therefore, the plagioclase







**Figure 10. *PT* diagrams and comp osition of phases in sample UM8. (a)** Equilibrium assemblage *PT* diagram (pseudosection) based on
the rock bulk rock composition reproducing well the observed primary assemblage GRT+CPX+Ky+Rt and the later formed crack-symplectite
assemblage CPX+OPX+ SPL+PL+ILM. The arrow denotes schematically the transition by isothermal decompression from the former to the
latter assemblage. **(b)** Isopleths of the modeled ternary garnet equilibrium composition. The shaded area corresponds to the stability field
of the primary assemblage. The yellow circle emphasizes the necessary features of a preserved equilibrium composition. The inlayed *PT*
diagram is an extension of figure (a) up to 20 kbar. **(c)** Implied isopleths of the measured GRT-types and their proximity to the stability field
of the primary assemblage denoted by the shaded area. The inlayed table summarizes the composition of the GRT-types. **(d)** Comparison of
the measured GRT-types, clinopyroxene and plagioclase inclusions with their calculated equilibrium mineral compositions in the range 15–6
kbar at 825 °C.





observed in inclusion domains in garnet is secondary. The calculated composition of plagioclase remains unchanged down to 6 kbar. Noticeably, the measured composition of plagioclase inclusions is in excellent agreement with the calculated one. The measured $X_{mg}$ and $X_{Al,T}$ of the clinopyroxene inclusions are very close to the calculated values. The clinopyroxene is *di*-rich and contains an appreciable amount of *cTs* component. In contrast, the measured $X_{Na}$ shows the largest deviation from its

calculated value pointing to almost complete loss of Na. This loss is probably related to the stabilization of plagioclase below ~12 kbar as well as to the formation of hornblende and the recrystallization of the matrix after decompression. This view is corroborated by the fact that the composition of the clinopyroxene inclusions are very similar ot the compositions of the clinopyroxene in the rock matrix.

The calculated results for the crack-symplectite assemblage CPX+OPX+SPL+PL, seed more light and reliability to the

*PT* evolution of the rock. Fig. 11a is an equilibrium assemblage diagram (pseudosection) calculated for a bulk composition equaling the initial garnet analysis #974 shown in Fig. 5e. A minor amount of quartz was added to this composition in order to suppress the formation of olivine implied by the excess $SiO_2$ of the bulk symplectite over the initial garnet, see Fig. 9 and section 5. This quartz is quickly consumed already during the first calculated steps of decompression shown in Fig. 11b. The stability field of the target assemblage CPX+OPX+SPL+PL in Fig. 11a refines the magnitude of decompression down to

pressures between 7.5 and 5 kbar and temperatures above 730 °C.

In Fig. 11b, the calculated mineral compositions are compared with those measured within the symplectite. Again, the pressure range 6–15 kbar at 825 °C is thought to be representative for the decompression. It is interesting to note that the measured initial garnet composition (a GRT-type C) lies close to the calculated garnet composition at, say 15 kbar. The diagram predicts that during decompression the garnet equilibrium composition remains fairly stable down to pressures of ~9 kbar. Then, it

changes continuously until its terminal composition at ~7.8 kbar has been reached. The agreement between the compositions of the contact and terminal garnet is particularly good, especially in case of Ca and Fe. This corroborates the view that, in contrast to clinopyroxene and plagioclase, garnet did not suffer further modification by post-decompression recrystallization. The large discrepancy between the measured and the calculated clinopyroxene compositions is due to the Na-free bulk composition used in the calculation. Accordingly, the calculated plagioclase in equilibrium with the phases shown in Fig. 11b is pure

anorthite, instead of the observed plagioclase with $X_{an} = 0.95$ (Supplement, Table S5).

Beside the contact garnet composition, the applied thermodynamic model reproduces very well also the measured modal distribution of the symplectitic phases (Table 3).

The above results are not biased by any external thermobarometric references. They are based solely on the bulk composition of the rock or on the initial garnet that were input to the applied thermodynamic model. The overall non-equilibrium state of

the garnet has been established not only by the Ca-distribution map shown in Fig. 2g, but also by checking, if the *grs*, *alm* and *prp* isopleths are crossing each other at a single point, providing evidence of preserved equilibrium (Fig. 10c). The calculated equilibrium assemblage diagrams reproduce well the observed assemblages and, in case of the isochemical break-down of garnet to the crack- symplectite, the measured composition and volumes of garnet and symplectitic phases.



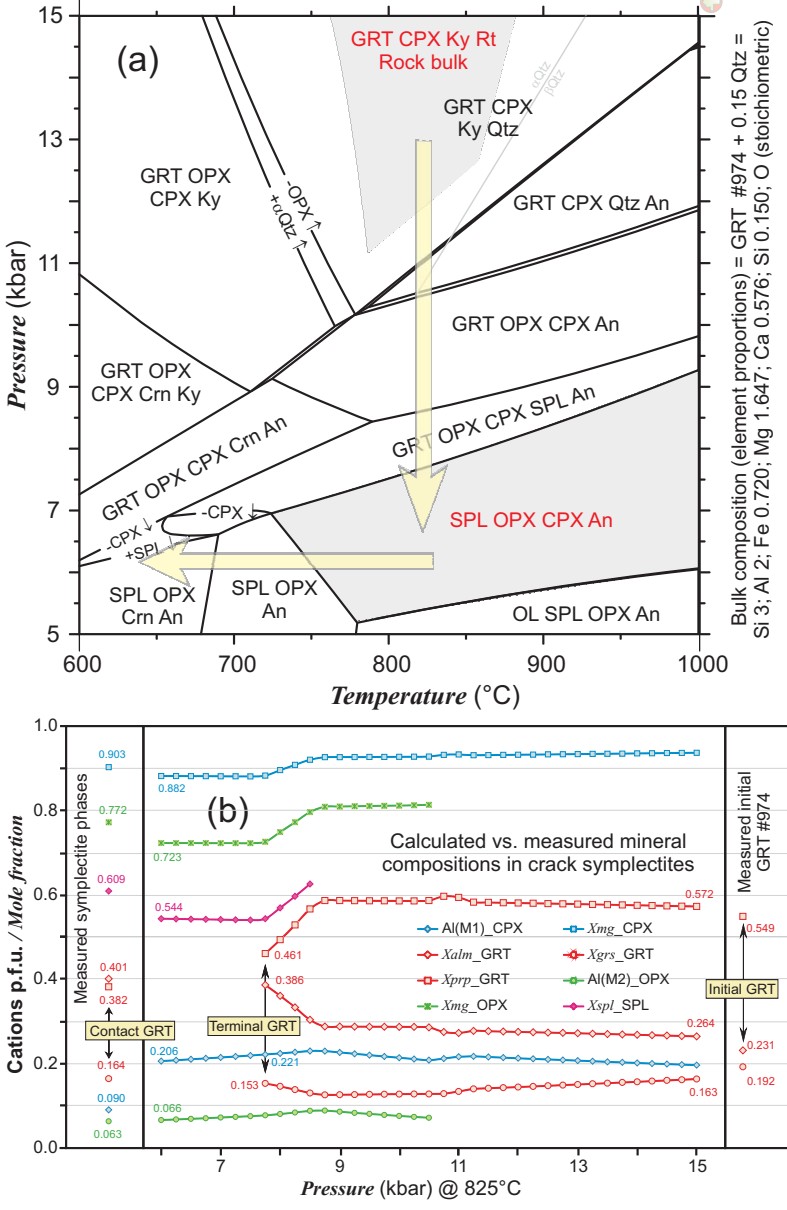

**Figure 11. Isochemical break-down of garnet to crack- symplectite.** (a) Equilibrium assemblage *PT* diagram (pseudosection) for a bulk composition equaling the composition of the initial garnet shown in Fig. 5e. The upper shaded area corresponds to the stability field of the primary assemblage GRT+CPX+Ky in Fig. 10a. The lower shaded area corresponds to the refined stability field of the observed crack-symplectite CPX+OPX+SPL+PL shown in Fig. 4d. The decompression arrow shows schematically the transition from the former to the latter assemblage. For the cooling arrow, see discussion in section 7.3. (b) Calculated mineral equilibrium composition along the decompression denoted by the arrow in figure (a). The terminal garnet is the last calculated composition before complete garnet elimination. The contact garnet is the spatially best resolved analysis of the retreating garnet edge in contact with symplectitic phases. Measured compositions of the initial garnet and symplectitic pyroxenes and spinel (cf. Fig. 5e) are shown for comparison. The results are summarized in Table 3.





**Table 3. Calculated vs. measured composition and volume of clinopyroxene, orthopyroxene, spinel and plagioclase building up crack-symplectites.** The calculated compositions and volumes are based on a bulk composition equaling the initial garnet in Fig. 5e. Methods of volumetric analysis and calculation of bulk symplectite composition are described in the Supplement.

| Phase | | Composition | | Volume-% | |
|---|---|---|---|---|---|
| | | Calculated @ 825°C, 7.75 kbar | Measured | Calculated @ 825°C, 7.75 kbar | Measured |
| CPX | $Al_{(T)}$ | 0.221 | 0.090 | 0.7 | 47.4 [a] |
| | $X_{mg}$ | 0.882 | 0.903 | | |
| | $X_{Na}$ | | 0.021 | | |
| OPX | $Al_{(M1)}$ | 0.077 | 0.063 | 47.2 | |
| | $X_{mg}$ | 0.726 | 0.772 | | |
| SPL | $X_{mg}$ | 0.543 | 0.609 | 10.1 | 12.3 |
| PL | $X_{an}$ | 1.000 | 0.945 | 40.1 | 40.3 |
| GRT [c] | $X_{alm}$ | 0.386 | 0.394 [b] | 1.9 | 0.0 |
| | $X_{prp}$ | 0.461 | 0.414 | | |
| | $X_{grs}$ | 0.153 | 0.185 | | |

(a)  Value includes 8.1 % HBL considered as substitute for CPX.

(b)  Value includes $X_{sps}$ = 0.027.

(c)  GRT-values corresond to  terminal (calculated) and contact (measured) garnet.

## 7  Discussion

### 7.1  Symplectite formation

Despite the single *PT* history experienced by the rock, the observed symplectite diversity points to the influence of the local environment on their formation. The inclusion-related symplectites are the products of reaction of garnet with its kyanite

inclusions. The rim- symplectites are the products of reaction of the garnet rim with the rock matrix. In case of the crack-symplectites, it is just the garnet instability that has led to its partial break-down, leading to the presumption that the crack-symplectites are isochemical to the garnet. This presumption has been verified by measurement of volume and composition of the symplectitic phases as well as by the results of the applied thermodynamic modeling.

Symplectite formation requires three processes to occur simultaneously. Firstly, chemical components must be redistributed

at the reaction front, because the precursor garnet (being considered chemically homogeneous on the scale of the forming symplectite) is replaced by an intergrowth of the chemically distinct symplectite phases at a sharp reaction front. The redistribution of these components occurs by diffusion within the reaction front (Gaidies et al., 2017). Secondly, for the reaction front to propagate, bonds must be broken in the precursor garnet at the leading edge of the reaction front, and new bonds must be established at its trailing edge forming the symplectite phases. The bulk of this process is referred to as *interface reaction*. And

thirdly, new interfaces are formed separating the different symplectite phases.

Symplectite formation is initiated by a change in the environmental conditions of the rock. In case of the sample at hand, by decompression at a more or less constant high temperature. The garnet in contact with the newly forming symplectitic phases has a distinctly different composition from the breaking-down garnet. This composition difference sets up transient gradients in chemical potential of the garnet components that decay with time. This process implies a time-dependent change of the

composition of the breaking-down and retreating garnet edge that is accomplished by intra-crystalline diffusion. This change




in composition is accompanied by recrystallization. In case of the inclusion-related symplectites, this recrystallization has led to a garnet edge that overgrew its own breaking-down products (Fig. 3d,e). The measured diffusion profiles at the breaking-down and retreating garnet edge (Fig. 5) reflect the "*frozen-in*" state of the above mentioned transient gradients at some time past the very first formation of the symplectite phases. Therefore, they allow for the calculation of the time scale of symplectite

evolution. This is discussed in the next section.

It is often observed that the characteristic inter-granular spacing (and size) of the symplectite phases decreases with decreasing temperature of symplectite formation. Remmert et al. (2018) presented experimental evidence corroborating this view and argued that the characteristic spacing of symplectite phases is small at low temperatures, whereby symplectite formation tends to be diffusion-controlled. The characteristic spacing becomes larger at higher temperatures, whereby diffusion is more effi-

cient, and symplectite formation becomes interface-reaction-controlled. The systematic decrease of the characteristic spacing of the symplectite phases from the older, external portions to the younger internal portions of the symplectite is observed in the rim and crack symplectites described earlier (Fig. 4).

EBSD analysis revealed a crystallographic orientation relationship (COR) between orthopyroxene and spinel in the rim- and crack- symplectites (see Supplement for details). According to Obata (2011) and Obata and Ozawa (2011), CORs occur

in symplectites around garnet in ultramafic rocks have formed at temperatures higher than ~800 °C; they are missing in symplectites that have formed at much lower temperatures. In the samples at hand, the COR between orthopyroxene and spinel is observed in both the early coarse-grained and the late fine-grained portions. Therefore, the *T*-change monitored by the symplectites of this study is rather small.

In summary, the studied symplectites have been formed most probably during cooling of the rock, but still at high tempera-

ture. It is thus very likely that the onset of symplectite evolution coincides with the transition from the more or less isothermal rock decompression to a more or less isobaric cooling (cf. Fig. 11a). This stage of rock evolution is discussed further below.

## 7.2 Time scale of symplectite formation

The systematic composition changes of garnet towards the interfaces to the symplectite can be explained by re-equilibration of the garnet that was mediated by intra-crystalline cation diffusion. One of the best examples observed in the studied samples

refers to the crack symplectite shown in Fig. 4d and the associated garnet profiles in Fig. 5e. The typically low Mn content of the garnet increases slightly towards the garnet–symplectite interface. This is probably due to the fact that the symplectite assemblage cannot accommodate the Mn of the garnet, which is then "pushed back" into the reactant garnet, where it is passively enriched immediately ahead of the symplectite reaction front. The Ca content remains largely unaffected, whereas Fe is successively enriched and Mg successively depleted towards the garnet–symplectite interface. The inter-diffusion of Ca,

Mg, Fe and Mn in garnet was modeled using a 1D implicit finite difference method. The diffusion coefficients were taken from Chakraborty and Ganguly (1992). Fixed concentration boundary conditions were defined by the contact garnet closest to the garnet-symplectite interface ($X_{alm} = 0.37$; $X_{prp} = 0.41$; $X_{grs} = 0.19$) and the initial garnet #974 ($X_{alm} = 0.24$; $X_{prp} = 0.55$; $X_{grs} = 0.20$), see Fig. 5e and Supplement Table S2. Mn ($X_{sps}$) was treated as the dependent component. To minimize cutting effects, the shortest profile T-Y shown in both figures was selected. Fitting of the model curves to the measured point analyses in Fig.




5e) yielded time scales of 340, 20 and 1.5 years for the assumed temperatures of 700, 800 and 900 °C, respectively. On the geological time scale, this corresponds to an ephemeral event. Such short-term diffusion cannot be explained by cooling alone. As will be discussed below, interface reactions responsible for symplectite formation is substantially mediated by the presence of $H_2O$-fluids. A limited fluid availability of such fluids may inhibit the progress of symplectite formation.

## 5  7.3  Synthesis

The complex compositional structure of the garnet in Fig. 2g cannot be explained simply as a (partially) preserved growth or (partial) homogenization at high temperature conditions. Additionally and as shown by the measured isopleths, the composition of all garnet types deviates from equilibrium, precluding thus any reliable thermo-barometric estimate of their formation conditions. The "older" GRT-type C occupies the large interior part of the garnet; its composition shows the least deviation
from equilibrium, corresponding closest to the stability field of the primary assemblage GRT+CPX+Ky. The Ca-rich GRT-type Z3 occupies a wide zone parallel to the garnet rim showing the largest observed deviation from equilibrium. GRT-type Z1 is the latest garnet type showing the closest relation to the action of metasomatizing agents. It is developed along a garnet crack, intrudes irregularly the garnet interior and extends in a narrow zone along the garnet rim. Finally, the distribution of garnet types in Fig. 2g shows a relative timing of formation that can be summarized by Z1 < Z3 < Z2 ≤ C.

All these features provide convincing evidence that the garnet shown in Fig. 2g has undergone diffusion-aided metasomatic modification during the late stages of its evolution represented best by the GRT-types Z3 and Z1. Accordingly, the "younger" GRT-type Z3 may evolve from the "older" GRT-type C by removal of a total amount of ~12 mol-% of Fe + Mg (Table 1; Fig. 5a). Recalling that GRT-type Z3 also predates symplectite formation, we may conclude that the metasomatic alteration that formed GRT-type Z3 along the lower garnet margins in Fig. 2g took place under *PT* conditions not substantially different from
those of the primary assemblage. An obvious metasomatizing agent might have been an anatectic melt and/or its accompanying fluids derived from the adjacent felsic rocks (cf. Hasalová et al., 2008a). Interestingly, Štípská et al. (2014a, b) inferred anatectic melts and fluids capable of transforming eclogites embedded in felsic granulites of the Blanský les massif to intermediate garnet and pyroxene-bearing granulites at pressures of ~12 kbar, but higher temperatures of 950 °C.

The calculated equilibrium assemblage diagram in Fig. 10 constrains the lower pressure of 11 kbar at ~800 °C for the
stability of the primary assemblage GRT+CPX+Ky. Due to compositional modification of garnet and resetting of clinopyroxene inclusions, a reliable estimate of pressure is more than questionable. Nevertheless, these conditions are comparable to peak conditions of 14.5 ± 2 kbar and 870 ± 50 °C estimated by Johansson and Möller (1986) for the same primary assemblage in mafic rocks of similarly Mg-, Al- and Ca-rich composition from Roan, Western Gneiss Region, Norway. Incidentally, these Norwegian rocks have evolved during retrogression to an "intermediate-*P*" granulite comprising orthopyroxene, spinel,
anorthite, andesine, sapphirine and corundum (Johansson and Möller, 1986).

Associated with the latest GRT-type Z1 are the poikiloblastic garnets with their numerous plagioclase inclusions. As shown in Fig. 8, these garnets evolved continuously on or from pre-existing, inclusion-poor garnet. Texturally very similar poikiloblastic garnets with numerous plagioclase inclusions have been described by Racek et al. (2008) in garnet- and pyroxene-bearing granulites of the Sankt Leonhard granulite massif (their sample D454). As these garnets did not match well with the other





features observed in their samples, they have been called by these authors "enigmatic". Despite the chemical and paragenetic differences that are certainly controlled by bulk composition, the strange poikiloblastic and "enigmatic" garnets are pointing to a common process that is seemingly acting not only locally on the granulites of Sankt Leonhard or on those of the Dunkelsteiner Wald, but on a regional scale. As in case of the GRT-type Z3 above, the suspicion falls again on the metasomatic agents related

with anatectic melts and/or associated fluids. The anatectic nature of at least the quartzo-feldspatic rocks in the Gföhl Nappe System and Drosendorf Nappe System has been inferred by Petrakakis (1986, 1997), Racek et al. (2006), Hasalová et al. (2008a), Schulmann et al. (2008) and references therein. Metasomatic alteration mechanisms affecting minerals and rocks by anatectic melts and/or fluids seem to be complex and are currently not clear. However, as far as the poikiloblastic garnets of this study are considered, it should be recalled (cf. Fig. 8) that they pre-date formation of the crack-symplectites. Their formation

is, therefore, related most probably to activity of metasomatizing agents during the latest stage of garnet evolution.

The inferred more or less isothermal decompression starting in the stability field of the primary assemblage above 11 kbar and ending in that of the crack-symplectite at 5–7 kbar (cf. Fig. 11) constrains also any other symplectite formed by break-down of the primary assemblage, e.g. also the inclusion-related symplectites not treated here in depth. For the very similar crack- and rim- symplectites, the appropriate reactions started until the pressure reached values of less than ~7 kbar. This

decompression was most probably related with the nearly vertical extrusion of the rocks to mid-crustal levels suggested by Štípská et al. (2004), Schulmann et al. (2005, 2008), Franěk et al. (2006), Tajčmanová et al. (2006), Racek et al. (2006), Duretz et al. (2011). This major deformation event was most likely responsible also for the development of subgrains within GRT-Type Z1 (cf. Fig. 7), and the opening of cracks within garnet (Fig. 3e) and rock matrix (Fig. 2a). We interpret the straight, darker, hornblende- and white mica-rich bands cross-cutting the rock matrix as cracks that have been healed during late, post-

decompression recrystallization conditions (see below). At the high temperatures indicated by the metamorphic assemblages of the sample at hand, the opening of such brittle structures implies rather high strain rates. The cracks in garnet and rock matrix have enabled fluid infiltration and favored component mobility to sites of interface reaction forming the symplectites.

On the other hand, the excellent preservation of the delicate symplectite microstructures, especially of the "unprotected" rim kelyphites, points to symplectite formation under low differential stress. This is supported also by the extensive matrix

recrystallization leading to the observed granoblastic matrix as well as by the complete absence of localized shearing even along favorable preexisting, brittle planar elements like the healed cracks shown in Fig. 2a (Mancktelow, 2008; Mancktelow et al., 2013, cf.). The crack- symplectites were formed largely at post-decompression conditions during isobaric cooling at ~6 kbar (cf. Fig. 11). As already discussed in section 7.1, symplectite formation during cooling is indicated by the micro-structural features shown by the symplectitic phases. Isothermal decompression of the Gföhl Nappe System followed by isobaric cooling

at similar pressures has been demonstrated also in GRT+OPX-granulites (Petrakakis and Jawecki, 1995; Petrakakis, 1997). This stage of the rock evolution is likely related with the lateral distribution of the rocks by horizontal channel flow within the middle-crust, as suggested by the authors cited in the previous paragraph.

It has been argued by Remmert et al. (2018) that symplectite formation only occurs in a rather narrow window of bulk water contents. Whereas the formation of symplectites is hampered in completely dry systems, a Garben microstructure with little

spatial organization is formed at high water contents. The presence of a fluid phase during formation of the crack- symplectites



is corroborated by the fact that within the crack- symplectites, which are largely isochemical with GRT-type C, the plagioclase contains some Na (cf. Fig. 9 and Supplement Table S5). This Na could not have been derived from the precursor garnet, but necessarily supplied from the environment. Very likely, at least some Na (and Si) has been transferred to the reaction front, and some Ca (and Al) in the opposite direction. The latter has possibly contributed to the *an*-enrichment of the outer rims of matrix

plagioclase (cf. Fig. 2c, Fig. 4a and b). The transfer of the chemical components most likely occurred by diffusion through fluid films at the grain boundaries and along triple junction arrays in the rock matrix. Furthermore, small volumes of HBL+PL-symplectite evolving from OPX+SP+PL- symplectite during the latest stages of kelyphite evolution provide additional direct evidence for the presence of a hydrous fluid. The observed slope inversion of the diffusion profile shown in Fig. 5d most likely reflects this change of the symplectitic assemblage. At least part of the fluid mediating symplectite formation at post-

decompression conditions may have been derived by the crystallization of anatectic melts. In this respect, the cracks in the rock matrix might have promoted fluid mobility by forming favorable pathways to reaction sites.

Despite the cooling documented by the symplectite microstructures, symplectite formation led to a very limited replacement of the garnet. This is also supported by the remarkably very short duration of symplectite formation calculated from the diffusion profile shown in Fig. 5e. Furthermore, the recrystallized matrix contains currently a high temperature, barely retrogressed

assemblage with abundant clinopyroxene and plagioclase and very minor hornblende. These features are consistent with limited availability of $H_2O$ rather, than with increasingly sluggish reaction kinetics induced by cooling. The reason for this might have been the healing of the rock cracks due to the enhanced local formation of hornblende and white mica.

Finally, the following question deserves some discussion. If metasomatic modification of FeO, MgO, CaO in garnet has taken place, how reliable are the presented equilibrium assemblage diagrams that rely on compositions assumed to be closed

for such system components? As the metasomatic modification has affected at least the garnet, but before decompression, the question applies only to the primary assemblage GRT+CPX+Ky. As discussed already, the modification of garnet seems to be minor and of the order of c. 12 mol-% of Ca equivalents. On the other hand, which could have been the alternative primary assemblage for this peraluminous, potassium-poor and CPX-rich rock beside GRT+CPX+Ky? Due to the small number of coexisting phases (cf. Fig. 10a), the high variance of the primary assemblage is related to its large stability field extending far

above 15 kbar (cf. inlay in Fig. 10b). This underlines its low sensitivity in terms of changing *PT* as well as other external factors like the chemical potentials of CaO, FeO and MgO involved in the metasomatic change of the garnet. These changes may have induced the observed displacements of composition in garnet away from equilibrium, but hardly the assemblage. Therefore and despite this metasomatic modification, we consider GRT+CPX+Ky as the primary assemblage, precluding, however, the derivation of reliable geobarometric estimates therefrom.

**8   Conclusions**

- Peraluminous, mafic granulites associated with the ultramafic rocks of the Dunkelsteiner Wald in the Gföhl Nappe System, Austria, contain large garnets embedded in a well recrystallized granoblastic matrix of clinopyroxene, plagioclase and minor hornblende. The garnets contain inclusions of kyanite and Al-rich, Na-poor clinopyroxene. They are partly



replaced by (a) symplectites around Ky-inclusions composed of Crn, SPR, SPL, PL, (b) symplectites at their rims comprising OPX, SPL, PL and minor HBL, and (c) symplectites within cracks containing OPX, CPX, SPL, PL and minor HBL. The diversity of these symplectites is locally controlled.

- Extensive microprobe analysis and element mapping revealed that the garnets are Ca- and Mg-rich and compositionally complex. They are built up from different composition patterns (GRT-types) showing a specific relation among Fe, Mg and Ca. These GRT-types may be repeatedly observed in garnets. The mode of occurrence and the cross-cutting relations among these GRT-types within large garnet crystals allowed for the derivation of their relative timing of formation. At least the latest GRT-types show strong evidence of metasomatic alteration. Related with these garnet types are peculiar poikiloblastic garnets with numerous plagioclase inclusions.

- Crack symplectites are shown to be largely isochemical with the oldest garnet type hosting them.

- Garnet in contact with symplectites shows a pronounced, diffusion-driven composition change within a couple of tens of $\mu$m ahead the garnet–symplectite interface. This change is linked to symplectite formation and has overprinted the underlying GRT-types. Therefore, symplectite formation post-dates the bulk compositional structure of garnet.

- Calculated equilibrium assemblage diagrams and analysis of the measured garnet component isopleths showed the following. (a) The primary assemblage GRT + Al-rich-CPX + Ky has been formed at pressures above 11 kbar and temperatures about 800 °C. (b) None of the measured GRT-types and compositions is a preserved equilibrium composition. (c) Especially the most Ca-rich GRT-types are affected by metasomatic loss of Fe and Mg. This alteration has taken place at *PT*- conditions close to the stability conditions of the primary assemblage. (d) The crack-symplectite assemblage is stable at pressures of 5–7 kbar and temperatures above ~730 °C.

- The rocks have undergone a more or less isothermal decompression to pressures between 5 and 7 kbar related to the extrusion of the rocks of the Gföhl Nappe System to middle-crustal levels. This major deformation event has led to the formation of subgrains in garnet as well as cracks in garnet and rock matrix.

- The decompression induced the formation of the symplectites. At least the crack- and rim- symplectites were formed after decompression within a short time interval of lesser than 0.5 ka during almost isobaric cooling at ~6 kbar under conditions and low differential stress. Symplectite formation was promoted by hydrous fluids.

- The limited extend of garnet replacement by symplectites, the very short time interval of symplectite formation and the preservation of a barely retrogressed, granoblastic rock matrix of clinopyroxene, plagioclase and minor hornblende indicates limited availability of $H_2O$ during cooling of the rocks.

- The applied thermodynamic model reproduces well the observed assemblages and the measured compositions and modal abundances of the symplectitic phases.

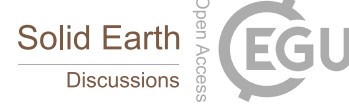

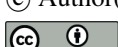

*Competing interests.*   The authors declare that they have no conflict of interest.

*Acknowledgements.*   Theo Ntaflos (Department of Lithosperic Research, University of Vienna) is acknowledged for providing the samples many years ago. Franz Kiraly (ibidem) is acknowledged for provision of technical expertise and assistance during microprobe work. Ben Huet and Christoph Iglseder (Geological Survey of Austria) have contributed to the goals of this paper with provision of geological maps,
5   critical comments and fruitful discussions.




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
