# Peer review of "Ca-rich garnets and associated symplectites in mafic peraluminous granulites from the Gföhl Nappe System, Austria"

_Solid Earth, 2018_

## Referee Comment (RC1) · H.-J. Massonne (Referee) · 9 Apr 2018

**Review of the manuscript "Ca-rich garnets and associated symplectites in mafic peraluminous granulites from the Gföhl Nappe System, Austria" by K. Petrakakis et al., submitted to Solid Earth**

General comments: The authors present a petrological study on a mafic granulite from the southeastern Bohemian Massif. The study of this rock is very detailed as the authors considered also, for example, the various symplectites in the granulite and the application of modern thermodynamic modeling techniques. Because of that the derived exhumation path in terms of pressure-temperature conditions is well documented. Thus, previous works proposing high peak-temperatures in the range around 1000°C for the granulites of the Gföhl unit seem to have overestimated these temperatures.
In summary, I would like to see the manuscript published soon after minor revisions.

Specific comments:
page 3, line 12: batholith?
page 5, lines 6-7: "but at much lower pressures of >4.5 kbar" - I am not sure which pressure range is addressed by the authors. My suggestion: "but at pressures between 4.5 and 6.5 kbar".
page 13, line 10: kelyphitized?
page 15, line 20: "GRT-type extends" - which garnet type?
page 18, lines 13-14: "does a common intersection point ... exists, indication that none of the GRT-types corresponds to a preserved equilibrium state" - I agree that this statement is likely, but not compelling. But other reasons are also possible and should be mentioned here. Perhaps the selected solid-solution models (here especially garnet) could be not fully adequate. The authors did not test the selected models and did not achieve their pseudosection calculations with alternative solid-solution models. A further reason for the missing fit of the isopleths could be the used bulk-rock composition although I also think that it is very likely that this composition has not changed during metamorphism.
page 18, line 19: "some prograde" - better: "a prograde"
page 20, line 9: "+PL, seed more" - probably: "+PL shed more"
page 22, line 5: "of reaction" - rather "of reactions"
page 22, line 16: something is missing at the end of the line - my suggestion "at hand, these conditions are characterized by"
page 23, line 4: "they allow for the calculation" - better: "they allow us to calculate"
page 23, line 6: "It is often observed" - better: "It is frequently suggested"
page 23, line 12: Replace "earlier" by "above"
page 23, line 15: "in ultramafic rocks have" - " in ultramafic rocks that have"
page 23, line 23: replace "composition" by "compositional"
page 23, line 27: "the Mn" - "the entire Mn"
page 24, line 4: "A limited fluid availability" - "A limited availability"
page 24, line 6: "zonation" might be better than "structure"
page 24, line 9: "conditions." - "conditions using pseudosections" or do the authors mean "thermo-barometric estimate"? If yes, I would not agree with as the size of the lamellae delivered (rough) temperature constraints.
page 24, line 10: "stability field" - better "P-T field" as the stability of the mentioned assemblage is more extended.
page 24, line 12: "It is developed" - better: "It formed"
page 24, lines 18-20: Seems to me somewhat speculative. This should be mentioned.
page 24, line 26: "is more than questionable" - "is questionable"

page 25, line 9: "pre-date formation" - "pre-date the formation"
page 25, line 10: "to activity" - "to the activity"
page 25, line 13: "in depth" - "in detail"
page 25, lines 15 and 31: "related with" - "related to"
page 26, line 16: "rather, than with increasingly" - "rather than increasingly"

Hans-Joachim Massonne (Universitaet Stuttgart)

---

## Author Comment (AC1) · 14 Apr 2018

Dear Mr. Massonne!

Thank you very much for the encouraging review. Your critical comments contribute to the quality and clarity of the paper, and are acknowledged accordingly.

In the following pages you will find your comments given in black color. My response to every comment is given in blue color. The text of the paper is given in two columns, italic typeface and blue color. The original text is on the left column, the modified one on the right.

I hope, I have responded satisfactorily to your critical comments.

With best regards

K. Petrakakis

**Review of the manuscript "Ca-rich garnets and associated symplectites in mafic peraluminous granulites from the Gföhl Nappe System, Austria" by K. Petrakakis et al., submitted to Solid Earth**

General comments: The authors present a petrological study on a mafic granulite from the southeastern Bohemian Massif. The study of this rock is very detailed as the authors considered also, for example, the various symplectites in the granulite and the application of modern thermodynamic modeling techniques. Because of that the derived exhumation path in terms of pressure-temperature conditions is well documented. Thus, previous works proposing high peak-temperatures in the range around 1000°C for the granulites of the Gföhl unit seem to have overestimated these temperatures.

In summary, I would like to see the manuscript published soon after minor revisions.

Specific comments:

page 3, line 12: batholith?

*No, it is wrong; replaced with "Pluton", thank you.*

page 5, lines 6-7: "but at much lower pressures of >4.5 kbar" - I am not sure which pressure range is addressed by the authors. My suggestion: "but at pressures between 4.5 and 6.5 kbar".

*The original formulation is indeed not clear, thank you. It is replaced as shown below.*

| | |
|---|---|
| *The Ostrong Nappe System differs distinctly from the other Moldanubian nappe systems. It shows similar anatectic temperatures of c. 720 °C, but at much lower pressures of >4.5 kbar. Based on rare relics of kyanite and staurolite, the rocks had attained a prograde maximum at <600 °C and c. 6 kbar before anatexis (Linner, 1996).* | *The Ostrong Nappe System is similarly anatectic, but differs distinctly in terms of pressure from the other Moldanubian nappe systems (Linner, 1996). Based on rare relics of kyanite and staurolite, the rocks had attained a prograde maximum at temperatures <600 °C and pressures of ~6 kbar. Subsequent anatexis took place at ~720°C and pressures not less than 4.5 kbar.* |

page 13, line 10: kelyphitized?

*My wrong formulation! Sentence replaced as shown below.*

| | |
|---|---|
| *It is noteworthy that the whole garnet is keliphitized along the rim (cf. Fig. 2g) and, ….* | *It is noteworthy that the garnet is replaced by keliphite along its rim (cf. Fig. 2g) and, ….* |

page 15, line 20: "GRT-type extends" - which garnet type?

*My sloppiness! Reformulated and (hopefully) better explained as shown below.*

*Modal analysis of the phases in crack-symplectites (Fig. 4d, Fig. 5e) combined with their chemical composition showed that the bulk symplectite composition is almost isochemical with the GRT-type C. GRT-type extends beyond the diffusion profile shown in Fig. 5e acquired over a crack-symplectite. An analysis practically identical with GRT-type C and lying on this profile is \#974 (Supplement, Table S3).*

*Garnet profiles acquired over crack-symplectites (e.g. Fig. 5e) show that GRT-type C extends beyond the diffusion-affected part of the garnet. An analysis practically identical with GRT-type C and lying on this profile is #974 (Supplement, Table S3). Modal analysis of the phases in crack-symplectites (Fig. 4d, Fig. 5e) combined with their chemical composition obtained by microprobe analysis showed that the bulk symplectite composition is almost isochemical with the GRT-type C.*

page 18, lines 13-14: "does a common intersection point ... exists, indication that none of the GRT-types corresponds to a preserved equilibrium state" - I agree that this statement is likely, but not compelling. But other reasons are also possible and should be mentioned here. Perhaps the selected solid-solution models (here especially garnet) could be not fully adequate. The authors did not test the selected models and did not achieve their pseudosection calculations with alternative solid-solution models. A further reason for the missing fit of the isopleths could be the used bulk-rock composition although I also think that it is very likely that this composition has not changed during metamorphism.

I fully agree with you that the absence of common intersection points of the garnet isopleths may be an artefact of the standard thermodynamic data and / or mixing models used. Let me follow you suggestion and focus on garnet.

The following figure is a re-calculation of the paper diagrams (Fig. 10) based on the widely used garnet data in the Thermocalc dataset  tcds55_p07.

The two target assemblages GRT+CPX+Ky (primary assemblage) and CPX+OXP+SPL+PL (in crack-symplectites) are well reproduced (cf. Fig. 10). Expectedly, the calculated isopleths are somehow different. Nevertheless, the measured isopleths lack again any common intersection points for all GRT-types.

Now, it can be argued that also this garnet model is not fully adequate. This might be the case. I think that every model has its own problems. But, on the other hand, the usefulness and reliability of a model can be tested by its ability to reproduce the observed assemblages and microstructures, and the measured compositions of the minerals (not necessarily by calculated P and T that may be suitable or non-suitable to personal prepossessions). With the applied thermodynamic, the outcomes of the submitted paper model (cf. Fig. 10) are based not on thermo-barometric determinations of P and T, which imply equilibrium compositions of the involved phases, but on the reproduction of the target assemblages and, in case of the crack-symplectites, mineral compositions (Fig. 11 and Table 3). This is discussed also on page 20, lines 29- 34. The composition of CPX in the primary assemblage cannot be reproduced, because it has been reset. Such a resetting of CPX is frequently observed. Similarly, the garnet composition cannot be fully reproduced, because it has been modified, presumably by intra-crystalline diffusion and metasomatic alteration. Only GRT-type C shows a composition closest to an equilibrium composition pertaining to the low-pressure part of the stability field of the primary assemblage.

[Figure]

Bulk= SI(16.697)TI(0.057)AL(9.543)FE(1.058)MG(4.875)
CA(6.336)NA(0.894)O(?)

Used garnet standard thermodynamic data and
mixing model GT07W2 from the Thermocalc
database tcds55_p07, as compiled for
Theriak/Domino by Douglas K. Tinkham
(http://dtinkham.net/index.html).

DKT is acknowledged for this precious work.

So, is the formulation in submitted paper justified? Taking account of your critical comment, I changed the
formulation as follows.

*Based on this figure* [Fig. 10b] *, Fig. 10c shows
the isopleths distribution for the observed GRT-
types. For none of the GRT-types does a
common intersection point of the isopleths for
the measured Xalm, Xprp and Xgrs exists,
indicating that none of the GRT-types
corresponds to a preserved equilibrium state.*

*Based on this figure* [Fig. 10b] *, Fig. 10c shows
the isopleths distribution for the observed GRT-
types. According to the applied thermodynamic
model, for none of the GRT-types does a
common intersection point of the isopleths for
the measured Xalm, Xprp and Xgrs exist. We
consider this as an indication of perturbation of
garnet composition from equilibrium.*

page 18, line 19: "some prograde" - better: "a prograde"

Yes, it is better. Replaced as suggested.

page 20, line 9: "+PL, seed more" - probably: "+PL shed more"

Yes, it is better, thank you. Replaced as suggested.

page 22, line 5: "of reaction" - rather "of reactions"

Yes, it is better, thank you. Replaced as suggested.

page 22, line 16: something is missing at the end of the line - my suggestion "at hand, these conditions are characterized by"

Thank you. Replaced with minor modification of your suggestion as shown below.

*Symplectite formation is initiated by a change in the environmental conditions of the rock. In case of the sample at hand, by decompression at a more or less constant high temperature.*

*Symplectite formation is initiated by a change in the environmental conditions of the rock. In case of the sample at hand, this change is characterized by decompression at a more or less constant high temperature.*

page 23, line 4: "they allow for the calculation" - better: "they allow us to calculate"

Replaced as suggested.

page 23, line 6: "It is often observed" - better: "It is frequently suggested"

Replaced with minor modification of your suggestion as shown.

*It is often observed that the characteristic inter-granular spacing (and size) of the symplectite phases decreases with decreasing temperature of symplectite formation. Remmert et al. (2018) presented experimental evidence corroborating this view and argued that the characteristic spacing of symplectite phases is small at low temperatures,                   ….*

*It is suggested that the frequently observed change of characteristic inter-granular spacing (and size) of the symplectite phases correlates positively with the temperature of symplectite formation. Remmert et al. (2018) presented experimental evidence corroborating this view and argued that the characteristic spacing of symplectite phases is small at low temperatures, ….*

page 23, line 12: Replace "earlier" by "above"

Replaced as suggested.

page 23, line 15: "in ultramafic rocks have" - " in ultramafic rocks that have"

Missing word inserted as suggested.

page 23, line 23: replace "composition" by "compositional"

Replaced as suggested.

page 23, line 27: "the Mn" - "the entire Mn"

Word "entire" inserted as suggested.

page 24, line 4: "A limited fluid availability" - "A limited availability"

Word "fluid" deleted as suggested.

page 24, line 6: "zonation" might be better than "structure"

Well, it is a matter of taste. I think, for most of the people, "zoning" or "zonation" is intuitively associated with "co-centric" element distribution patterns, for example, as those related to growth zoning. As this is not the case in the measured garnets and in order to avoid possible misunderstandings, I have chosen the phrase: "The complex compositional structure of the garnet".

page 24, line 9: "conditions." - "conditions using pseudosections" or do the authors mean "thermo-barometric estimate"? If yes, I would not agree with as the size of the lamellae delivered (rough) temperature constraints.

Well, I don't quite understand the point here, especially in respect to "as the size of the lamellae delivered (rough) temperature constraints". I may guess, your comment may be related with the discussion above. Therefore, the formulation (page 24, lines 7-11) is slightly changed as follows.

*Additionally and as shown by the measured isopleths, the composition of all garnet types deviates from equilibrium, precluding thus any reliable thermo-barometric estimate of their formation conditions. The "older" GRT-type C occupies the large interior part of the garnet; its composition shows the least deviation from equilibrium, corresponding closest to the stability field of the primary assemblage GRT+CPX+Ky*

*.*

*Additionally and as shown by the measured isopleths, the composition of all garnet types seems to deviate from equilibrium. The "older" GRT-type C occupies the large interior part of the garnet; its composition shows the least deviation from equilibrium, corresponding closest to PT conditions pertaining to the stability field of the primary assemblage GRT+CPX+Ky.*

page 24, line 10: "stability field" - better "P-T field" as the stability of the mentioned assemblage is more extended.

Right, thank you! The necessary change in line 10

>    *…, corresponding closest to PT-conditions pertaining to the stability field of the primary assemblage GRT+CPX+Ky.*

is already included in the previous reformulation.

page 24, line 12: "It is developed" - better: "It formed"

Replaced as suggested.

page 24, lines 18-20: Seems to me somewhat speculative. This should be mentioned.

We are trying here to give an answer to the question, when did the metasomatic alteration take place along the PT-path of the rock. We argue as following.

The "younger" GRT-type Z3 (cf. Ca-distribution map, Fig. 2g) might have been evolved from the "older" GRT-type C by removal of Fe+Mg. This process should have taken place before symplectite formation, because diffusion profiles have overprinted this GRT-type (Fig. 5). Additionally, the "youngest" GRT-type Z1 (replacing GRT-type Z3, see the Ca-distribution map, Fig. 2g) shows intracrystalline deformation that, as explained further below in the text, is related to the tectonically induced decompression. This argument is added in the new formulation below. So, the answer to the question _may_ be the following: The metasomatic modification should have taken place before decompression, i.e. under conditions most probably close to those of the primary assemblage. All these features comprise a line of argumentation or suggestion (or speculation?) with emphasis given by the used formulations "may evolve", "we may conclude", "may have taken place". Taking account of your critical comment, the new formulation (of course without underlining) is as follows.

*All these features provide convincing evidence that the garnet shown in Fig. 2g has undergone diffusion-aided metasomatic modification during the late stages of its evolution represented best by the GRT-types Z3 and Z1. Accordingly, the "younger" GRT-type Z3 may evolve from the "older" GRT-type C by removal of a total amount of ~12 mol-% of Fe + Mg (Table 1; Fig. 5a). Recalling that GRT-type Z3 also predates symplectite formation, we may conclude that the metasomatic alteration that formed GRT-type Z3 along the lower garnet margins in Fig. 2g took place under PT conditions not substantially different from those of the primary assemblage.*

*All these features provide convincing evidence that the garnet shown in Fig. 2g has undergone diffusion-aided metasomatic modification during the late stages of its evolution represented best by GRT-types Z3 and Z1. Accordingly, the "younger" GRT-type Z3 might have been evolved from the "older" GRT-type C by removal of a total amount of ~12 mol-% of Fe + Mg (Table 1; Fig. 5a). Recalling that GRT-type Z3 predates symplectite formation and that the GRT-type Z1 shows intracrystalline deformation (Fig. 7) due to tectonic displacement (see below), we may conclude that the metasomatic alteration that formed GRT-type Z3 and Z1 may have taken place under PT conditions close to the formation conditions of the primary assemblage.*

page 24, line 26: "is more than questionable" - "is questionable"

Changed as suggested.

page 25, line 9: "pre-date formation" - "pre-date the formation"

Changed as suggested.

page 25, line 10: "to activity" - "to the activity"

Changed as suggested.

page 25, line 13: "in depth" - "in detail"

Changed as suggested.

page 25, lines 15 and 31: "related with" - "related to"

Both changed as suggested.

page 26, line 16: "rather, than with increasingly" - "rather than increasingly"

Changed as suggested.

Hans-Joachim Massonne (Universitaet Stuttgart)

---

## Referee Comment (RC2) · M. Racek (Referee) · 16 Apr 2018

**The referee comment to the manuscript "Ca-rich garnets and associated symplectites in mafic peraluminous granulites from the Gföhl Nappe System, Austria" by authors Konstantin Petrakakis et al.**

**Genral Comments:**

The reviewed manuscript "Ca-rich garnets and associated symplectites in mafic peraluminous granulites from the Gföhl Nappe System, Austria" by authors Konstantin Petrakakis et al. represents a very detailed and focussed study of processes recorded in mafic granulites from Bohemian Massif mainly as a complex zoning of garnets.

The work is of a particular interest from both the more regional point of view, but mainly for a broader audience due to its quite unconventional approach, which seems to be quite appropriate for a study of such peculiar lithologies. The manuscript fits well within the scope of SE, is contains large set of new data and includes novel approach for estimates on metamorphic history of high-grade rocks. The conclusions are reached by relevant and clearly outlined methods and are fully justified. The methods are described and explained in detail allowing anybody to reproduce them. The substantial part of the manuscript is new authors contribution, while any references to previous works of other authors are properly cited. The title is relevant to the manuscript content, abstract summarizes the most important information reached by the work. The presentation is generally well structured, except of some minor flaws in the descriptive part making it a bit hard to follow (see specific comments). The language is fluent without any obvious mistakes (as far as I can recognize not being native speaker). Symbols and abbreviations are properly defined and used, references and supplementary material are appropriate. There are only several rather minor issues mostly of formal character, where I would recommend to make some changes in structure of some figures and text - mainly descriptive part. Also, I have few comments regarding the section about thermodynamical modelling - see specific comments.

Overall, it can be summarized as follows - scientific significance - excellent, scientific quality - excellent, presentation quality - good to excellent.

In conclusion, I recommend the manuscript to be accepted after rather minor revisions.

**Specific Comments**

Page 3 - lines 4 - 5: it is stated that "three lithotectonic nappe systems are generally dipping to the east" - however I have impression that the general structure of the eastern marging of the Moldanubian zone is gently west-dipping.

Page 6 - line 11 (and elsewhere) - here stated "as will be discussed later, its origin is secondary". In general, I would prefer that the description and interpretations of the features would be more separated - first the description, interpretation later. This is not the case of this manuscript and in some cases (as this), the interpretative statements are incorporated in the descriptive text without some supportive argument.

Page 7 - Figure 2: The Figure 2G seems not to be on appropriate place. It is not cited between Fig 2F and Fig 3. Since it is a figure showing already features of mineral chemistry, the reader does

not have information to understand all the indicated garnet types etc. that are shown in this figure. I think it should be somehow involved in Fig 5.

Page 11 - Figure 5C: The diffusion profile X-Y is asymmetrical. While the left (X) part has features described in the text (decrease of Ca etc.), the right side (Y) is missing them and in fact, the trends of the zoning resemble those observed by the profiles E-F, Q-S, and T-U (although the zoning is much less pronounced). Maybe this could be discussed in the text?

Page 12 - lines 6 - 7. You mention that the garnet C and Z1 have similar compositional characteristic, but you distinguish them based on the character of their occurrence. With respect to the Fig 2G - can you exclude that the C and Z1 garnets are really not the same type, taking into consideration possible effects of section of the garnet?

Page 13 - line 6 - 7. Sentence "GRT-type Z1 ... metasomatizing agents" is pure interpretation, however the reader does not yet have any background to understand it.

Page 14 - line 8: thin exsolution lamellae - of which phase?

Page 17 - line 9: statement "is isochemical to GRT-type C" should be rather "is almost isochemical". I think that this is not just a small detail but important point that although the local bulk rock chemistry did not deviate too much from the original garnet composition, the involvement of fluid (or melt) and the subsequent minor change of the bulk rock chemistry is crucial and the symplectites would probably not form without it.

Page 17 - line 17: It is worth to mention that the a-x model of cpx including the CaTs substitution was recently developed by Green et al. (2016 - Activity–-composition relations for the calculation of partial melting equilibria in metabasic rocks. Journal of Metamorphic Geology). I can only speculate, what would be the impact of using this model (together with the recent datset 6 by Holland and Powell 2011 and adequate a-x models of other solid solutions) on the calculated mineral chemistry. I know that this would need to be done by using either thermocalc or perple_x, but anyway, I think it is worth to test it.

Page 22 - Table 3: The table shows good agreement of the calculated and measured composition and modal proportion of phases. However I don't understand why the measured pyroxenes are not divided to CPX and OPX? (I would expect based on the presented images that the measured proportion of CPX would be considerably higher than the calculated one).

Page 22 - lines 5 - 8. See the comment Page 17 - line 17. I recommend to reformulate the sentence. The symplectites are not completely isochemical, as it is illustrated by the Fig 9.

Page 24 - lines 15 - 16. It is not clear if authors suppose that the garnet zoning was developed during the garnet growth or by modification of already existing garnet by diffusion during the metasomatism.

**Technical Corrections**

Page 2  - line 6 - Jedlička (diacritics)

- line 7 cf. Table 1 (missing space)

- from line 28 - list of abbreviations - missing abbreviations for muscovite and prehnite that are used later

Page 7 - Figure 2 - B - labels of minerals would be helpful. C - lines C-D and A-B are not explained in figure caption, as well as the elipse.

Pages 7, 9, and 10 - Figures 2, 3, and 4. The format if the figures is not unified. The scales and labels have often various fonts (see Fig 3C) and font size, sometimes are bold (see Fig 3B). The scales by BSE images sometimes involve information about voltage and current, sometimes not (see Figure 4). It would very good to unify the format of all the figures.

Page 9 - Figure 3: 3A - try to avoid intersection of line (arrow) wit text. Check formate of thearrow pointing from 3A to 3D. 3B - should be SPR and PL (instead of Spr and Pl). 3D - muscovite and prehnite are missing in the list of abbreviations. 3G - what is the strange bright rectangle in the centre of the image? Generally, there is not much visible in this figure.

Page 10 - Figure 4D + caption - profile T-Y is probably the profile T-U in the Fig 2G. Please check the Y/U throughout the text.

Page 11 - Figure 5: Maybe it would be helpful to mark the exact limits of described garnet types in the profiles by some vertical lines? 5D - I cannot find the E-F profile marked in any BSE of OM image.

Page 13 - line 11: Fig 5A - should be Fig 2G? I can see no dotted line in Fig 5A.

Page 19 - Figure 10B: Even after reading the figure caption, I am not sure what the yellow circle stands for. Should it just generally symbolize intersection of isopleths?

Martin Racek

Institute of Petrology and Structural Geology

Faculty of Science, Charles University, Prague

---

## Author Comment (AC2) · 20 Apr 2018

Dear Mr. Racek!

It is an honor for me and all co-authors that you, a "Moldanubian specialist", have evaluated positively the submitted paper. Of special value are your critical comments, the exact reading and inspection of the figures. You have contributed to the clarity and value of the paper and this is acknowledged by us accordingly.

In the following pages you will find your comments given in black color. My response to every comment is given in blue color. The text of the paper is given in two columns, italic typeface and blue color. The original text is on the left column, the modified one on the right.

I hope, I have responded satisfactorily to your critical comments.

With best regards

K. Petrakakis

**The referee comment to the manuscript "Ca-rich garnets and associated symplectites in mafic peraluminous granulites from the Gföhl Nappe System, Austria" by authors Konstantin Petrakakis et al.**

**Genral Comments:**

The reviewed manuscript "Ca-rich garnets and associated symplectites in mafic peraluminous granulites from the Gföhl Nappe System, Austria" by authors Konstantin Petrakakis et al. represents a very detailed and focussed study of processes recorded in mafic granulites from Bohemian Massif mainly as a complex zoning of garnets.

The work is of a particular interest from both the more regional point of view, but mainly for a broader audience due to its quite unconventional approach, which seems to be quite appropriate for a study of such peculiar lithologies. The manuscript fits well within the scope of SE, is contains large set of new data and includes novel approach for estimates on metamorphic history of high-grade rocks. The conclusions are reached by relevant and clearly outlined methods and are fully justified. The methods are described and explained in detail allowing anybody to reproduce them. The substantial part of the manuscript is new authors contribution, while any references to previous works of other authors are properly cited. The title is relevant to the manuscript content, abstract summarizes the most important information reached by the work. The presentation is generally well structured, except of some minor flaws in the descriptive part making it a bit hard to follow (see specific comments). The language is fluent without any obvious mistakes (as far as I can recognize not being native speaker). Symbols and abbreviations are properly defined and used, references and supplementary material are appropriate. There are only several rather minor issues mostly of formal character, where I would recommend to make some changes in structure of some figures and text - mainly descriptive part. Also, I have few comments regarding the section about thermodynamical modelling - see specific comments.

Overall, it can be summarized as follows - scientific significance - excellent, scientific quality - excellent, presentation quality - good to excellent.

In conclusion, I recommend the manuscript to be accepted after rather minor revisions.

**Specific Comments**

Page 3 - lines 4 - 5: it is stated that "three lithotectonic nappe systems are generally dipping to the east" - however I have impression that the general structure of the eastern marging of the Moldanubian zone is gently west-dipping.

Considering Fig. 1 in your excellent work on the Drosendoerf Window (Racek et al. 2006), I fully understand this remark. However, "Generally dipping to the east" is meant as follows. In the Austrian part of Moldanulbia the "traditional" succession of lithotectonic units from west to east is the following (works of Fuchs, Matura etc.). Ostrong Unit (tectonically lowest) / Drosendorf Unit (tectonically intermediate) / Gföhl Unit (tectonically highest). This implies that the general / regional dip is to the east (according to these authors related to the so called "Intramoldanubischer Deckenbau"). This general relation may have been modified on a local scale during the multi-stage tectonic evolution of

Moldanubia, especially during the lateral spreading of the units. As stated in Racek et al. (2006), (p. 228 and caption Fig. 9) the westward dip in the Drosendorf Window is the result of reworking of the earlier subvertical S2 during the latest deformation D3. D3 has been induced by the indentation of the Brunia basement.

Page 6 - line 11 (and elsewhere) - here stated "as will be discussed later, its origin is secondary". In general, I would prefer that the description and interpretations of the features would be more separated - first the description, interpretation later. This is not the case of this manuscript and in some cases (as this), the interpretative statements are incorporated in the descriptive text without some supportive argument.

I see the point and I have reduced such "occurrences" in case this reduction was appropriate. In some cases, it was not possible. For example, in the section Mineral Chemistry. There, the relative age relations between the garnets types are derived first. What follows in this section is about the pronounced diffusion profiles that overprint the garnet types. Implicit with this overprinting is the relative age of symplectite formation, if and only if the link between diffusion profile and symplectite formation is established. Let's see the text there, page 12, line 11

> *Such compositional zoning profiles within the reactive and retreating garnet edge are imposed over pre-existing garnet compositional patterns and are, therefore, secondary. As will be discussed later, their evolution is linked to the formation of the symplectites.*

Further below on page 13, line 7

> *As such overprinting relations have not been observed in case of the younger GRT-type Z1, we conclude that symplectite formation is at earliest coeval to this garnet type and consequently younger than the internal compositional structure of the garnet related to GRT-types Z2, Z3 and C.*

Without the necessary link, a large part of this section should be moved to section "Discussion". Similarly any other occurrence of such diffusion profiles and the resulting age relations, for example the garnets with subgrains and diffusion profiles, or the poikiloblastic garnet with diffusion profiles etc. should be moved accordingly. In my opinion, this would be inappropriate.

Page 7 - Figure 2: The Figure 2G seems not to be on appropriate place. It is not cited between Fig 2F and Fig 3. Since it is a figure showing already features of mineral chemistry, the reader does not have information to understand all the indicated garnet types etc. that are shown in this figure. I think it should be somehow involved in Fig 5.

I fully understand this remark. As the number of figures / templates is rather large, it took me a lot of time to arrange them in a way that saves space. But, as you say, the result is not the best one. I think, as Fig. 5 contains too many profiles, Fig. 2g cannot be incorporated there. I changed the  arrangement of the figure and the numbering by subtracting Fig 2g from Fig. 2, "making" it Fig. 6 and re-numbering the subsequent figures. It sounds easy, but resizing Fig. 2 and adding Fig. 6 resulted in a new pagination of the paper by LaTeX. Therefore, a certain text part originally on page X, line y may now be displaced to

page A, line b. Therefore, your suggested text changes are discussed below with reference to the old and the new pagination.

Page 11 - Figure 5C: The diffusion profile X-Y is asymmetrical. While the left (X) part has features described in the text (decrease of Ca etc.), the right side (Y) is missing them and in fact, the trends of the zoning resemble those observed by the profiles E-F, Q-S, and T-U (although the zoning is much less pronounced). Maybe this could be discussed in the text?

Strictly geometrically, yes, the right part (Y) is not a mirror image of the left part (X), because 3 analyses at the beginning of part Y have been rejected. The reason was that their quality was not OK due to bad local polishing at the garnet/symplectite interface. Should I have discussed this? Nevertheless, I think, the key point is that the component trends are "symmetrical" and, additionally, similar with those in profile P1-P2. Profiles X-Y and P1-P2 are crossing inclusion-related symplectites. I don't agree that they resemble profiles E-F, Q-S, T-U. Therefore, all these different profiles are put in Fig. 5, else they would be redundant. The differences between these profiles were described on page 13, lines 23 – 30. As I have used the formulations "the former" and the "latter" by referring to two figures too often, I changed the text accordingly to make it (hopefully) clearer. Please, see the changes in the original text marked in yellow color below.

Page 13, lines 23 - 30

Page 14, line 1 –9

*The garnet profiles shown in Figs. 5c,e are acquired over domains of inclusion-related and crack-symplectites, respectively. The former shows a diffusion profile within about 10 $\square m$ towards the garnet–symplectite interface characterized by increasing Xprp and Xalm, and sharply decreasing Xgrs. The latter diffusion profile is characterized by unchanged Xgrs, increasing Xalm and decreasing Xprp. This striking difference reflects primarily the influence of the local environment on their formation mechanism. In the former case, the environment is defined by GRT-type Z2 reacting with its kyanite inclusions. In the latter case, it is defined by the instability of GRT-type C alone. Diffusion profiles at rim- symplectites of GRT-type C as the one in Fig. 5d are similar to those of crack-symplectites. Occasionally and as shown in Fig. 5d, the diffusion curves for Mg, Fe, Mn adjacent to rim symplectites show an inflection point at some short distance before the retreating garnet edge.*

*The garnet profiles shown in Fig. 5c and Fig. 5e are acquired over domains of inclusion-related and crack-symplectites, respectively. Fig. 5c shows a diffusion profile within about 10 $\square m$ towards the garnet–symplectite interface characterized by increasing Xprp and Xalm, and sharply decreasing Xgrs. In Fig. 5e, the diffusion profile is characterized by unchanged Xgrs, increasing Xalm and decreasing Xprp. This striking difference reflects primarily the influence of the local environment on their formation mechanism. In case of Fig. 5c, the environment is defined by GRT-type Z2 reacting with its kyanite inclusions. In case of Fig. 5e, it is defined solely by the instability of GRT-type C. Diffusion profiles at garnet-rim symplectites are similar to those at crack-symplectites. Occasionally, and as shown in Fig. 5d, the diffusion curves for Mg, Fe, Mn adjacent to rim symplectites show an inflection point at some short distance before the retreating garnet edge.*

Page 12 - lines 6 - 7. You mention that the garnet C and Z1 have similar compositional characteristic, but you distinguish them based on the character of their occurrence. With respect to the Fig 2G - can you exclude that the C and Z1 garnets are really not the same type, taking into consideration possible effects of section of the garnet?

Let me start with the latter part of your question regarding possible cutting effects. The garnet in Fig 2g is the largest observed within the collection of all samples. Therefore, I believe, this section "goes" through the garnet "center". Then I ask myself, how GRT-Type Z1, which

a) is related to a garnet crack (Figs 2G and Fig. 2A),
b) "intrudes" irregularly GRT-Type Z2 (Fig. 2G),
c) is related by this mode of occurrence with some mobile phase,
d) cross-cuts GRT-Type Z3 (Fig. 2G),
e) occurs at garnet margins (Fig. 2G)

can be related to GRT-Type C, which is spatially restricted in the homogenous (not "intruding" or so) inner/central, part of the garnet?

My answer and suggestion is, they are not related. These matters are discussed already in page 13, lines 1 – 8 (now page 9, line 33 to page 12, line 6). Let me copy this text here.

> *GRT-type C occupies the large, inclusion-poor interior part of the garnet. GRT-type Z1 has evolved at a strongly Ca- depleted area along a garnet crack, which can be recognized in Fig. 2a,g. Therefrom, it "intrudes" irregularly the garnet interior and extends over a narrow zone along the lowest rim of the garnet (Fig. 5g). As can be recognized in Fig. 2g, GRT-type Z1 cross-cuts type Z3 over a narrow transitional zone and is therefore younger. This age relation is supported also by the typical middle-sized garnet in Fig. 5b. This garnet is of type Z3, but has evolved to GRT-type Z1 towards its margin. Compared with the other GRT-types shown in Fig. 2g, GRT-type Z1 is a late feature related most probably with the action of metasomatizing agents. GRT-types Z2 and C are seemingly older, but their temporal interrelation is not clear. Their transition towards GRT-type Z3 is smooth.*

However, your remark has given me the opportunity to re-evaluate the way I have formulated this "similarity". Thank you. It is probably better to emphasize that this similarity consists of sharing the same relation Xgrs < Xalm < Xprp, which is not shared by the other GRT-Types, see Table 2. So, I changed the text as follows.

| Page 12, line 6. | Page 9, line 24 |
|---|---|
| *GRT-types Z1 and C show similar compositions characterized by Xgrs < Xalm < Xprp. However, they differ distinctly in their mode of occurrence, see below.* | *GRT-types Z1 and C share the same component relation characterized by Xgrs < Xalm < Xprp.* |

Page 13, lines 8 – 9.

*Despite the compositional similarities, we discriminate GRT-type C from GRT-type Z1 based on their different modes of occurrence.*

Page 12, lines 6-7

*Despite their similar component relation described earlier (Xgrs < Xalm < Xprp), we discriminate GRT-type C from GRT-type Z1 based on their strikingly different modes of occurrence described above.*

Page 13 - line 6 - 7. Sentence "GRT-type Z1 … metasomatizing agents" is pure interpretation, however the reader does not yet have any background to understand it.

I fully agree, yes, it is an interpretation based on the criteria listed in the previous discussion and fulfilled by GRT-type Z1, see the copied original text above. I am asking myself,

'What background a reader of Solid Earth might need in order to understand that this GRT-Type (Ca-depleted, crack-related, "intruding" the other garnet types, etc.) can be interpreted as a product of metasomatic modification'

and I can't find an answer.

Page 14 - line 8: thin exsolution lamellae - of which phase?

Thank you for this remark. In fact, the lamellae are very thin and could not be resolved by the microprobe analysis. Therefore, the text is slightly changed as follows.

Page 14, line 8

*As shown in Fig. 2c, the interiors of some larger matrix clinopyroxene crystals contain very thin exsolution lamellae.*

Page 15, lines 2-3

*As shown in Fig. 2c, the interiors of some larger matrix clinopyroxene crystals contain very thin, analytically unresolved exsolution lamellae.*

Page 17 - line 9: statement "is isochemical to GRT-type C" should be rather "is almost isochemical". I think that this is not just a small detail but important point that although the local bulk rock chemistry did not deviate too much from the original garnet composition, the involvement of fluid (or melt) and the subsequent minor change of the bulk rock chemistry is crucial and the symplectites would probably not form without it.

Right, your formulation is better, thank you. It has already been changed. It is also appropriate in view of the fact that, indeed, the deviation from "pure isochemical" is already discussed a few lines above.

Page 17, line 9

*The second one is the crack- symplectite assemblage CPX+OPX+SPL+PL that is isochemical to GRT-type C.*

Page 17, lines 16-17

*The second one is the crack- symplectite assemblage CPX+OPX+SPL+PL that is almost isochemical to GRT-type C.*

Page 17 - line 17: It is worth to mention that the a-x model of cpx including the CaTs substitution was recently developed by Green et al. (2016 - Activity–-composition relations for the calculation of partial melting equilibria in metabasic rocks. Journal of Metamorphic Geology). I can only speculate, what would be the impact of using this model (together with the recent datset 6 by Holland and Powell 2011 and adequate a-x models of other solid solutions) on the calculated mineral chemistry. I know that this would need to be done by using either thermocalc or perple_x, but anyway, I think it is worth to test it.

Please appreciate that as a "very, very, very small" co-author of the Theriak/Domino paper (that is based on a huge, long-lasting work of Chrstian De Capitani), I rather prefer to operate with this. Your idea to test various datasets is interesting in a very general sense, but not the purpose of this paper. As you ascertain below, the used model "worked" good by reproducing the observed assemblages and the measured mineral compositions. In my opinion, this is the crucial point. Please, take also a notice of the ongoing discussion with Mr. Massonne (first reviewer) about a similar topic. So, I try to keep things as simple as possible as long as they reproduce observation.

Page 22 - Table 3: The table shows good agreement of the calculated and measured composition and modal proportion of phases. However I don't understand why the measured pyroxenes are not divided to CPX and OPX? (I would expect based on the presented images that the measured proportion of CPX would be considerably higher than the calculated one).

Thank you for the reliability confirmation of the applied model. The distinction between CPX and OPX during volumetric analysis was not possible, because both phases showed the same gray color in the used BSE-images. This information and in general the methods used are explained in the Supplement, page 2, line 35 to page 3, line 55. The reader is invited in the caption of Table 3 to get all relevant information therefrom.

Page 22 - lines 5 - 8. See the comment Page 17 - line 17. I recommend to reformulate the sentence. The symplectites are not completely isochemical, as it is illustrated by the Fig 9.

Right, it is reformulated as shown below. However, please note that this sentence is talking about a "presumption" implied by the observed microstructural features.

Page 22, lines 5 - 8

*In case of the crack-symplectites, it is just the garnet instability that has led to its partial break-down, leading to the presumption that the crack- symplectites are isochemical to the garnet.*

Page 22, lines 23-25

*In case of the crack-symplectites, it is the fact that garnet was less stable and broke-down partially, leading thus to the presumption that the crack- symplectites are more or less isochemical to the garnet.*

Page 24 - lines 15 - 16. It is not clear if authors suppose that the garnet zoning was developed during the garnet growth or by modification of already existing garnet by diffusion during the metasomatism.

Let me copy the original text here and underline the key-word.

> *"All these features provide convincing evidence that the garnet shown in Fig. 2g has undergone diffusion-aided metasomatic modification during the late stages of its evolution represented best by GRT-types Z3 and Z1."*

I think, it is clear that the current compositional structure is a metasomatically induced modification, not a product of growth or homogenization or both combined. This implies certainly a pre-existing garnet at the time of modification. How this pre-existing garnet did look like, I cannot say with certainty and without risking a huge degree of speculation. Two limiting cases are, however, possible. Typical growth zoning and typical homogenization by intracrystalline diffusion due to high T. The true state before modification might have been somewhere between these two limiting cases.

Please note that I avoid the formulation "zoning" or "zonation" for this garnet, as these words may connote to the reader "growth zoning". Therefore, I prefer the formulation "compositional structure" to describe a pre-existing garnet (with growth zoning and/or homogenization and/or both) that has been asymmetrically modified (GRT-types not concentrically distributed, cross-cutting relations etc.) by metasomatic action. Please see also my response to Mr. Massonne about this topic.

**Technical Corrections**

Page 2 - line 6 - Jedlička (diacritics)

Done, thank you!

- line 7 cf. Table 1 (missing space)

Done, thank you!

- from line 28 - list of abbreviations - missing abbreviations for muscovite and prehnite that are used later

Thank you for this remark. Yes, the abbreviations Ms and Prh are used (e.g. in Fig 2), but unfortunately not explained. This is already corrected.

Page 7 - Figure 2 - B - labels of minerals would be helpful. C - lines C-D and A-B are not explained in figure caption, as well as the elipse.

Thank you for the constructive suggestions. Labels in Fig. 2b inserted. Profile A-B in Fig. 2c is not used and, therefore, the line is deleted. Profile C-D is shown in Fig. 6B. So, the caption looks now as follows.

Page 7 - Figure 2 – B

*(c)* *BSE image of the rock matrix showing smooth interfaces and triple junctions among clinopyroxene, plagioclase and hornblende as well as thin exsolution lamellas in the clinopyroxene interior. The plagioclase rims are enriched with Ca.*

Page 7 – Caption Figure 2b

*(c)* *BSE image of the rock matrix showing smooth interfaces and triple junctions among clinopyroxene, plagioclase and hornblende as well as thin exsolution lamellas in the clinopyroxene interior designated with an ellipsis. The plagioclase rims are enriched with Ca. The profile C-D is shown in Fig. 7b.*

Dear Mr. Racek!

You are a perfectionistic, exact observer. Thank you. To your three remarks below, I say nothing more, than, please, see Fig. 3 as an example of the re-edited figures according to your suggestions. It was a little bit hard, but it helped!

[Figure]

Pages 7, 9, and 10 - Figures 2, 3, and 4. The format if the figures is not unified. The scales and labels have often various fonts (see Fig 3C) and font size, sometimes are bold (see Fig 3B). The scales by BSE images sometimes involve information about voltage and current, sometimes not (see Figure 4). It would very good to unify the format of all the figures.

Page 9 - Figure 3: 3A - try to avoid intersection of line (arrow) wit text. Check formate of the arrow pointing from 3A to 3D. 3B - should be SPR and PL (instead of Spr and Pl). 3D - muscovite and prehnite are missing in the list of abbreviations. 3G - what is the strange bright rectangle in the centre of the image? Generally, there is not much visible in this figure.

Page 10 - Figure 4D + caption - profile T-Y is probably the profile T-U in the Fig 2G. Please check the Y/U throughout the text.

Page 11 - Figure 5: Maybe it would be helpful to mark the exact limits of described garnet types in the profiles by some vertical lines? 5D - I cannot find the E-F profile marked in any BSE of OM image.

*I see the point, but, I think that Fig. 5 is overloaded with symbols, boxes and text. However, the caption in Table 2, where the average compositions with standard deviations of for the GRT-Types are given, invites the reader to see the Supplement. There, in Fig. S 2, those parts of the large profile Z1-Z2-Z3 (615 point analyses!) are re-plotted, which were incorporated into average calculation. The plots there are at higher resolution and are intended to show that the averaged analyses are several consecutive analyses of nearly constant composition along the profile Z1-Z2-Z3.*

Page 13 - line 11: Fig 5A - should be Fig 2G? I can see no dotted line in Fig 5A.

*You are a perfect reader! Yes, Fig. 5a is wrong. The text is now changed as follows.*

Page 13 - line 11                                          Page 12, lines 8-10.

*It is noteworthy that the garnet is replaced by keliphite along its rim (cf. Fig. 2 g) and, as shown by the dotted line at the lowest garnet rim (Fig. 5a), only the younger GRT-type Z1 may be formed as late as the rim- symplectite.*

*It is noteworthy that the garnet is replaced by keliphite along its rim (cf. Fig. 2 g) and, as shown by the dotted line at the lowest garnet rim, only the younger GRT-type Z1 may be formed as late as the rim- symplectite.*

Page 19 - Figure 10B: Even after reading the figure caption, I am not sure what the yellow circle stands for. Should it just generally symbolize intersection of isopleths?

*Thanks, it is to emphasize, that the recognition of a preserved equilibrium composition requires a common intersection point of the isopleths. It is (hopefully) better formulated now, as follows.*

Page 19 - Figure 10B                                       Page 19 - Figure 10B

*The yellow circle emphasizes the necessary features of a preserved equilibrium composition.*

*The yellow circle emphasizes the necessity of a common intersection point of the isopleths in case of a preserved equilibrium composition.*

Martin Racek

Institute of Petrology and Structural Geology

Faculty of Science, Charles University, Prague

---

## Author Response (AR2)

**Dear Editor Mr. Rossetti!**

I want to express my sincere appreciation for your detailed analysis of our manuscript and the reviewer's comments. Your input certainly helped us to further improve the manuscript.

Some of your statements are, however, difficult to apprehend in the light of the referees' comments and the information given in the MS.

Let me first address this difficulty in a general sense.

(Henceforth, your comments are given in black color, italic typeface. My/our response is given in blue color. Underlining of parts of your comments is mine).

**Your comments §1a and §3 (first part):**

1a - the Abstract, Introduction and Discussion sections are too focused to the specific geological case that, despite interesting, should be better introduced to the broad audience of the journal. Apart the local significance, what is the general geological problem that the manuscript aims to address and characterise? which the (expected) advancement of knowledge in the discipline (tectonometamorphic evolution of high-grade basement terrains)?, etc.

(3) - Impact of the study (see also point 1a above). As it stands, despite providing interesting results, the manuscript appears as a regional study of local significance.

For the convenience of the reader following this discussion, let me copy/paste here the general comments of the referees (bold typeface is mine).

**Hans-Joachim Massonne:**

"I would like to see the manuscript published soon after minor revisions".

**Martin Racek:**

- "The work is of a particular interest from both the more regional point of view, but mainly for a broader audience due to its quite unconventional approach, which seems to be quite appropriate for a study of such peculiar lithologies"...
- "The manuscript ... contains large set of **new data** and **includes novel approach for estimates**" ...
- "The substantial part of the manuscript is new authors contribution" ...
- "Overall, it can be summarized as follows scientific significance excellent, scientific quality excellent, presentation quality good to excellent)".

Finally, let me add the statement given by you prior to your comments:

**Your revised version adequately addressed the reviewers' comments.**

As they stand, the reviewer's comments and your comments diverge significantly. If the MS is "too focused to the specific geological case" and the "general geological problem" as well as the "advancement of knowledge" are missing, and the MS is of "local significance", then revising a manuscript designed as a case study with a regional link to one with a more general significance is a

major task and certainly at odds with "minor revisions", as given in your statement just at the beginning of your comments:

*Topical Editor Decision: Publish subject to minor revisions (review by editor) (02 May 2018) by Federico Rossetti*

Even if this communication is not necessarily conclusive in itself, we see the point you raise, especially in the last part of

**your §3:**

The Authors are thus encouraged to expand the implications of this study and extract results that can be of interest for a broader audience, i.e. broaden the impact of the presented results away from the study area. A specific sub-section of the Discussion dedicated to these broad implications would greatly increase the impact and relevance of the study.

In this sense, we have modified and extended accordingly the sections "Abstract", "Introduction", "Discussion" and "Conclusions". Please, see these changes and extensions after our response to the specific points of your criticism.

Now let me turn specifically to your comments and questions.

The "general geological problem": In fact, we address a petrological problem in the Introduction (the way PT-conditions are commonly calculated), which has major geodynamic implications. The "advancement of knowledge" has been implicitly or explicitly emphasized by the referees. Please, see also our response to your comment §1c.

**Your §1b:**

The Introduction, rather than focusing just on the specific case, should instead introduce the general geological problem, its significance, gaps of knowledge and aims the manuscript pursues with respect to the existing background information. On this regard, Table 1 should be part of the geological background, rather than part of the Introduction section. In other words: Why this study should be of interest for a broad audience?

As mentioned above, the Introduction was/is setting up a petrologic problem with major geodynamic implications. Let me describe this problem shortly. The tectono-metamorphic evolution of the high-grade Moldanubian rocks is based almost solely on strongly divergent PT-estimates summarized in Table 1. These PT-estimates imply severe, contrasting, mostly speculative, geodynamic implications. The "gaps of knowledge" were/are stated too, namely calculating PT by using garnet, particularly the Ca- content and/or Ca-zoning, which is supposed to be invariably robust against modifications other than those implied by changing PT. The "aims the manuscript pursues" were/are given too, namely that garnets treated in the MS and particularly their Ca contents are shown to be susceptible to change also by intracrystalline diffusion and metasomatic processes, calling thus into question the above assumption. We offered/offer physically based criteria and methods for evaluating observed microstructures and composition patterns. These criteria are generally applicable and, we reckon, that they are of interest to a broad audience involved in evaluating petrographic data and in quantifying PT-conditions of rock metamorphism from metamorphic rocks themselves. It is rather boring to list here the numerous places of the MS, where implications beyond the regional/local

context are addressed. A short list of outcomes interesting for a broad audience is given in our response to the last underlined part of your §1c.

Table 1 is an integral part of the problem set up and "belongs" primarily to the Introduction. It emphasizes the divergence of PT-estimates calculated on the basis of the above assumption; it is complemented with geodynamic implications therefrom, which are mostly speculative. Of course, Table 1 serves also the Section "Geological background".

**Your §1c:**

*Criteria for sample selection, sample location and constituent mineralogy are not provided. A table is needed. As far I have understood, your study is based just on one sample? How representative? How chosen? This information would greatly help the reader to evaluate the scientific rationale followed in this study.*

There are many points "condensed" here. Let us start with the sample location and copy/paste page 5, lines 9 - 14 of the MS you have annotated, as well as Caption of Fig. 1 (the only change with regard to the version you have annotated, is the original word "inlay" replaced by "inset" according to your suggestions in the annotated pfd-file).

"Seven samples of mafic granulites were collected from **loose boulders dispersed over the steep flanks of the Mitterbachgraben** (inset, Fig. 1). The local bedrock comprises serpentinites pertaining to the mantle-derived peridotite of the Dunkelsteiner Wald. The ultramafic rocks form several 100 m long lensoid bodies embedded in mylonitic felsic granulite. The collected samples are noticeable in the field, as they do not belong to the regionally widespread rock types in this area. They are dark gray, middle to fine-grained, mostly granofelsic mafic granulites containing abundant pyroxene and kelyphitic reddishbrown garnets of occasionally striking large size up to 1.5 cm."

Figure 1. "Simplified geological map of the Austrian part of the Moldanubian Unit modified after Schnabel (2002), Krenmayr et al. (2006), Cháb et al. (2007) and Kalvoda et al. (2008). DW signifies the Drosendorf Window. **The inset is a sketch of the sampling area** of the mafic granulites (embedded within felsig granulites) at the **steep flanks of Mitterbachgraben**. **The land road Gansbach—Kicking as well as the GPS-coordinates at the star are shown** (after Sheet 37, "Mautern", Geological Survey of Austria)."

So, we think that the sampling location is adequately given. It is not about outcrops to be listed with GPS coordinates in a Table, but loose boulders at the steep flanks of the Mitterbachgraben, indeed, a sampling location of restricted areal extent.

The sampling criteria are also given; please see the underlined part of the text above. Additionally, the information is given (page 5, lines 15 - 20) that the collected rocks are interesting objects by previous work done on them (Carswell et al., 1989).

The sample description is given too. The samples are described chemically by stating their normative contents (page 5, lines 21-32) and compared, due to their peculiar composition, with other similar or related rocks known from literature. The rock analyses are given in the Supplement. Here is the appropriate text, please, re-evaluate.

XRF-analyses of the collected samples reveal K-poor, Mg-rich compositions with Xmg ranging within 0.70–0.82 (see Supplement). In terms of normative contents, they contain crn in the range 10 to 16 %. Three of them contain an in the range 9 to 19 %; the other four samples contain instead ol between 3 and 9 %. The di contents of all samples vary between 48 and 61% and the hy content between 3 and 19%. The ab content varies between 9 and 19%. Some of the samples (UM5, UM6, UM8) resemble the corundum-bearing garnet clinopyroxenites from the Beni Bousera ultramafic massif that have been considered as low pressure crystallization cumulates from plagioclase-rich gabbros of ophiolitic affinity that underwent subduction and re-equilibration at mantle conditions (Kornprobst et al., 1990). Svojtka et al. (2016) assigned the Dunkelsteiner Wald pyroxenites to LREE-enriched melts of the subcontinental lithospheric mantle. In comparison with Al2O3 contents of 15-24 wt.-% in the samples presented here, their pyroxenitic samples are characterized by significantly lower Al2O3 not exceeding 12.23 30 wt-% and higher Xmg = 0.87 - 0.90. Based on the pronounced peraluminous composition variability, we consider our samples as mantlederived clinopyroxenitic melts that have assimilated variable amounts of Al-rich crustal material during ascent and tectonic emplacement to their current position.

To your question about the one sample and its representative power. Yes, the MS is dealing with one sample and more sections thereof. All collected samples are mineralogically similar (Grt+CPX+PL with minor HbL), but the one selected for intensive investigation contains exceptionally large garnets with an exceptionally high number of features described in the MS. Please, re-evaluate page 5, lines 32 – stating the following.

The following presentation and discussion is focused on sample UM8 that contains some large garnets with an unusual high number of features. This sample is the most magnesian and peraluminous of the whole collection (Xmg = 0.82 with normative crn = 14.07 % and an = 18.78 %).

In order to avoid further misunderstandings, the above text has been extended in the new version as follows (the essential part is underlined):

The collected samples are mineralogically similar containing slightly variable amounts of clinopyroxene, plagioclase and garnet accompanied by some hornblende. The following presentation .....

Regarding the last underlined part of your §1c. I don't quite understand, what is meant here. I can only guess. Indeed, when collecting these samples, we could not anticipate that they will provoke questions about Ca-rich garnets, about their potential susceptibility to metasomatic modification and re-adjustment of composition by intracrystalline diffusion, about their almost isochemical breakdown to symplectites under "Moldanubian" conditions, about the recognition of their state of equilibrium or non-equilibrium, about reliability by using them in PT-calculations, about intragranular deformation features, about poikiloblast-resembling growth, about a lot of other features reported in the MS. We think that all these aspects, though obvious to a specific study object, are of interest in the broadest context of petrology and assessment of the tectono-metamorphic history of high-grade rocks (cf. about "advancement of knowledge" in your comment §1a). In fact, we are very happy and very lucky to own these samples, especially the one treated in the MS.

Your §2a:

A figure dealing with the regional geology is not provided in the Introductio section, despite essential for people not familiar with the regional geology.

This is somewhat puzzling. The appropriate Fig. 1 showing a simplified geological map is on page 4 and is referring to the second section of the MS dealing, in our opinion, adequately with the regional geology. That a figure or whatever "dealing with the regional geology" is part of the introduction is very new to me. According to your comment §1b

"The Introduction, rather than focusing just on the specific case, should instead introduce the general geological problem, ... "

**Your §2b:**

a Materials and Method section is missing and this information is mixed up with the introduction statements (see the ending part of the Introduction section). This section should also provide analytical details.

This is half of the truth. Indeed, such information is missing in the MS, but, as stated in page 2, lines 26-27 it is present in the Supplement. Why in the Supplement? We have always been concerned about the length of the paper and decided to put it there, because it takes another three full pages including an additional Table and an additional additional figure (of course tables of rock and microprobe analyses are excluded). The claimed information is not missing, but placed into the supplement. Please, re-evaluate the Supplement and appreciate the volume of data, the transparent and exact description of methods as well as the volume of work done on this single sample. Further three pages about devices, device settings, recalculation methods etc. may be boring for the general reader that wishes first to get the essential outcomes of the MS. For the reader interested in these "technical" matters, the Supplement provides thorough information. But, on the other hand, pointing to the Methods by reference to the Supplement at the last paragraph of the Introduction, is, indeed, not elegant. So, we created a short new section called "Methods of data acquisition and recalculation" providing essentially the same information about the Supplement together with mineral abbreviation etc. Please, check the new version.

**Your §3 (last part):**

The Authors are thus encouraged to expand the implications of this study and extract results that can be of interest for a broader audience, i.e. broaden the impact of the presented results away from the study area. A specific sub-section of the Discussion dedicated to these broad implications would greatly increase the impact and relevance of the study.

Yes, this is indeed a straightforward constructive point. Things can always be done better. In this sense, we have partly reformulated and / or extended the Abstract, the Introduction, the Discussion and the Conclusions. The new versions are given in the next pages.

**At the end of your comments:**

*I have provided an annotated version of the manuscript, where these and other points that need further consideration by the Authors are detailed.*

Thank you very much, it was very useful.

Broader impact of this study is primarily depending on how this criticism will be addressed in the revised version. Critical on this regard is the re-organisation of the Introduction, Discussion and Conclusion sections

We have done our best.

With best regards.

K. Petrakakis in the name of all co-authors

**New versions**

The most essential changes/insertions/extensions relative to the version you have annotated are underlined.

[revised manuscript text omitted]